# Estimating growth patterns and driver effects in tumor evolution from individual samples

Leonidas Salichos [1,2], William Meyerson[1,2], Jonathan Warrell [1,2] & Mark Gerstein [1,2,3,4]*

Tumors accumulate thousands of mutations, and sequencing them has given rise to methods for finding cancer drivers via mutational recurrence. However, these methods require large cohorts and underperform for low recurrence. Recently, ultra-deep sequencing has enabled accurate measurement of VAFs (variant-allele frequencies) for mutations, allowing the determination of evolutionary trajectories. Here, based solely on the VAF spectrum for an individual sample, we report on a method that identifies drivers and quantifies tumor growth. Drivers introduce perturbations into the spectrum, and our method uses the frequency of hitchhiking mutations preceding a driver to measure this. As validation, we use simulation models and 993 tumors from the Pan-Cancer Analysis of Whole Genomes (PCAWG) Consortium with previously identified drivers. Then we apply our method to an ultra-deep sequenced acute myeloid leukemia (AML) tumor and identify known cancer genes and additional driver candidates. In summary, our framework presents opportunities for personalized driver diagnosis using sequencing data from a single individual.

[1] Program in Computational Biology and Bioinformatics, Yale University, New Haven, CT 06520, USA. [2] Department of Molecular Biophysics and Biochemistry, Yale University, New Haven, CT 06520, USA. [3] Department of Computer Science, Yale University, New Haven, CT 06520, USA. [4] Center for Biomedical Data Science, Yale University, New Haven, CT 06520, USA. *email: pi@gersteinlab.org

Over the past several decades, researchers have proposed different models to explain tumor progression, including stochastic progression, the mutator phenotype, and clonal evolution[1–3]. Originally suggested about 40 years ago[3], Navin and colleagues provided strong evidence that the "punctuated clonal evolution" model constitutes a major force in cancer progression. According to this model, tumor progression is an evolving system subject to selective pressure while accumulating thousands of mutations[4,5].

Advances in technology have allowed scientists to sequence thousands of genomes, revealing millions of variants per individual[6–8]. In cancer genomics, The Cancer Genome Atlas (TCGA)[4] offers access to thousands of cases encompassing over 30 types of cancer. Similarly, the International Cancer Genome Consortium (ICGC) recently announced "data release 26", which comprises data from more than 17,000 cancer donors and 21 tumor sites. Within ICGC, the pancancer analysis of whole genomes (PCAWG) study is an international collaboration to identify common patterns of mutations in over 2800 sequenced whole-cancer genomes[9]. As cancer databases continue to expand, the amount of fully sequenced genomes will continue to increase, with future plans setting goals for the storage of more than a million genomes[10]. Concurrently, deeper sequencing signifies less noise, more accurate variant-allele frequencies (VAFs), and more accurate subclonal and single-nucleotide variant (SNV) identification, while increasing the detection of novel drivers[11–13].

Recent studies have tackled the effect of selection in tumor progression in the context of clonal evolution, neutral evolution, and selection, providing valuable insights about the clonal progression of the disease[5,14–16]. By considering tumor progression as an evolutionary process, cancer development follows the trajectory of different evolutionary pathways based on cell and population dynamics, optimization strategies, and selective forces. These evolutionary trajectories have been shown to influence primary tumor growth[17] and the timing of landmark events[18]. However, the evolutionary and selective mechanisms during tumor progression remain unexplored and strongly debated[19–21].

Accumulated SNVs have been characterized as drivers or passengers, depending on whether or not they provide a selective advantage for the tumor cells. If the selective advantage or their respective effect is weak, the mutations are known as mini-drivers, although the existence and detectability of mini-drivers has been debated[22,23]. Identifying SNV and gene drivers has been one of the focal points of cancer genomics, where different methods aim to detect driver mutations based on selection, recurrence or changes in mutational density[22,24]. These methods rely on the deviation from our expectation of the underlying genomic mutation rates, often by considering additional covariates such as replication timing and gene expression[25–27]. Other methods, characterized as ratiometric, assess the composition of mutations, normalized by the total mutations in a gene[22]. This includes the proportion of inactivating mutations, recurrent missense mutations, functional impact bias, mutational composition, or clustering patterns[28–31]. However, if only a small proportion of mutations within a genomic region (which is potentially under negative selection or functional restrictions) facilitates cancer progression, driver detection requires either a very large sample, a strong effect or otherwise the driver's presence is undetectable[22]. Further, mutational heterogeneity in cancer poses an additional problem for large cohorts; as the sample size increases, so does the list of putatively significant genes, producing many false-positive driver genes[26]. More importantly, only a minimal portion of driver mutations are, in fact, true drivers[32]. This is particularly important in a clinical context as assessing a cancer gene mutation as a true functional driver is a critical problem for drug selection[32,33].

According to recent studies[34] and in agreement with past theories[35], a few major genetic hits (strong drivers) can induce tumorigenesis. At the same time, a driver mutation may not actually be the cause of tumorigenesis, but instead only increase growth rate and therefore be under positive selection[36]. One of the most common and widely used lists of cancer genes is the Vogelstein list[28], consisting of ~140 oncogenes and tumor-suppressor genes (TSGs). While high-impact mutations in TSGs might favor cancer progression by deactivating tumor suppression, oncogenes need altered expression levels to favor tumor growth. Thus, high-impact mutations, such as nonsense mutations in oncogenes might decrease gene expression and burden tumor cells[37]. Less appreciated is the role of noncoding mutations in tumor progression[36,38,39]. Interestingly, in the case of TSGs, different studies have reported the role of noncoding intronic mutations that alter correct exon splicing, resulting in faulty tumor suppression[40–43]. Similarly, in the case of oncogenes different studies have reported the potential effect of synonymous mutations[39,40,44]. For example, Gartner et al.[44] showed that the early synonymous mutation F17F in the *BLC2*-like 12 gene alters the binding affinity of regulatory hsa-miR-671-5p, leading to changes in expression.

In our study, we developed a framework to model tumor progression and the effect of drivers in individual deep-sequenced tumors. We successfully applied our model using 993 linear tumors (linear subclonal expansion, where each parent subclone has one child subclone) from the PCAWG consortium, and found that predicted drivers[9] are associated with periods of positive growth. Our results suggest that mutations involved in biological processes such as cell development, cell differentiation, and multicellularity appear under strong positive or negative growth enrichment. Missense or nonsense mutations in TSGs were enriched during positive growth. We also identified significant positive enrichment for mutations in the promoter regions of both TSGs and oncogenes. In addition, in the case of TSGs, we discovered a small but significant signal from intronic mutations. Finally, we applied our framework to a deep-sequenced model acute myeloid leukemia (AML) tumor, where our predicted growth peaks aligned closely with three missense mutations from known cancer genes. Notably, our analysis suggests the potential presence of additional driver candidates.

## Results

**Method formalism.** When sequencing a cell population or tumor bulk, each mutation is assigned a variant-allele frequency (VAF), which corresponds to the mutation's frequency in the resulting pool. According to the infinite sites model[45], once a mutation occurs it will continue to exist within that cell and its descendants. Therefore, if we assume that there is no selection or chromosomal duplications, the VAF is associated with the time of occurrence and population growth rates. That is, in the presence of a driver (i.e., in cells with higher fitness), nondriver mutations within that cell lineage will also have higher-than-expected VAF and are termed "hitchhikers"[28] (Fig. 1 and Supplementary Notes). Hitchhikers that initially occurred before the driver mutation but continue to exist within that cell lineage will have a VAF that is higher than or equal to the driver's frequency. We call these hitchhikers "generational" (g-hitchhikers) because they essentially mark the different generations of an ever-increasing number of tumor cells and thus exhibit a clock-like behavior. Since any nondriver lineage derived from the division of earlier cells will result in a mutation having lower frequency, these predriver hitchhiking mutations will indicate generational growth (Fig. 1). As the fitness mutation becomes more prevalent over time, so

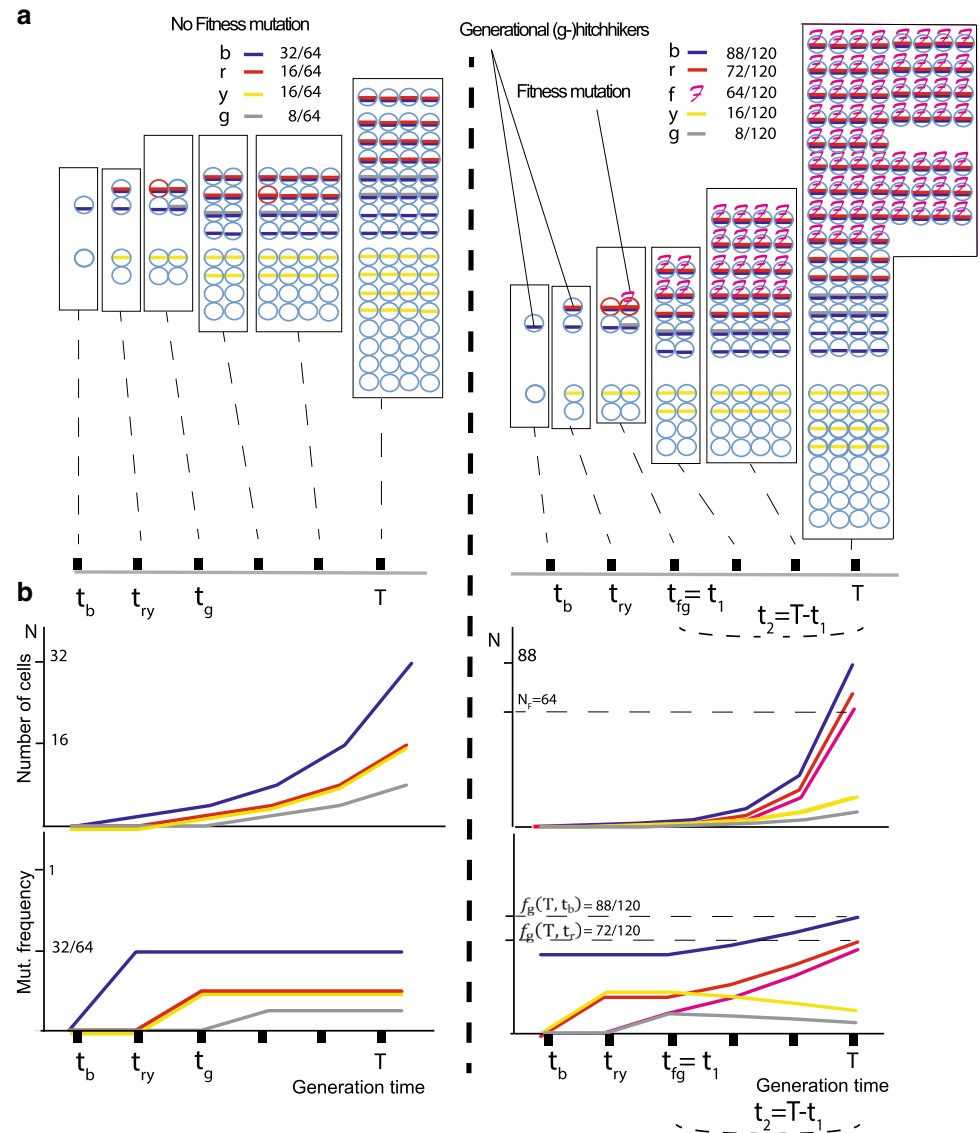

**Fig. 1 (g−)Hitchhikers' frequency depends on driver's effect.** We consider a simple population of cancer cells that grows exponentially $N(t) = e^{rt}$; for simplicity, we assign one mutation per cell division. At the time of biopsy $T$, the frequency of a mutation occurring at time $t_n$ would be equal to $f_n(T, t_n) = \frac{e^{r(T-t_n)}}{e^{rT}} = e^{-rt_n}$. At time $t_1$, a mutation occurs that increases the growth rate $r$ of the specific subpopulation by a scalar multiplier $k$, such that the new population is now expanding as $N_F = e^{krt_2}$. Thus, at the time of biopsy $T = t_1 + t_2$, we expect a generational (g−) hitchhiking mutation that occurred at time $t_m < t_1$ to have a frequency equal to $f_g(T, t_m) = \frac{e^{r(T-t_m)} + N_F - e^{rt_2}}{N_{tot}}$, where $N_{tot}$ is the total number of cells (or mutations) and $N_F$ is the number of cells that contain the fitness mutation that occurred at $t_1$ and expanded for $t_2$. Therefore $N_F = e^{krt_2}$. In **a**, we show the mutational frequencies at the time of biopsy $T$ for two growth models; one neutral and one with a fitness mutation occurring at time $t_1 = t_{fg}$. Hitchhiking mutations "b" (blue), "r" (red), as well as passenger mutations "g" (gray) and "y" (yellow), also occur at different time points. **b** Under an exponential model with a fitness mutation occurring at time $t_1 = t_{fg}$, hitchhikers "b" and "r" show an increased frequency compared to neutral, subject to time and effect of the fitness mutation. Passenger mutations "y" and "g" that occurred before or with the fitness mutation, but on a different cell lineage, end up with lower frequencies. We characterize mutations "b" and "r" as generational (g−) hitchhikers since they mark the population's generational growth.

does the prevalence of predriver g-hitchhikers, but critically at a different pace, which we calculate (see Methods).

Our framework's equations relate the VAF of generational hitchhiker mutations to the fitness effect of the subclonal driver with which they are hitchhiking, mediated by various growth and population parameters (i.e., the base growth rate $r$, a scalar multiplier $k$ corresponding to fitness effect of the mutation, the time $t_1$ when the driver mutation is generated, $N_{tot}$ the population size and $N_F$ the driver's subclone size). The existence and fitness effects of subclonal drivers are not directly observable but are of primary biomedical importance. The VAF of hitchhiker mutations is directly observable, therefore we chose to use these VAFs

to infer the presence of subclonal drivers and estimate their fitness effects. Our approach is to fit the known VAFs of the hitchhiker mutations in the hitchhiker equations to estimate the growth pattern and the fitness effect of subclonal drivers. This method requires to simultaneous estimate the various growth and population parameters, which we performed using nonlinear least-squares optimization. To address the fact that real tumors differ from idealized behavior, we make use of sliding windows and local timepoint reoptimizations in the parameter estimation to prevent departures from idealized behavior in one part of the VAF spectrum from interfering with parameter estimation in other parts of the VAF spectrum. We derived our estimators

for $r$ and $k$ through the implementation of a deterministic model to a stochastic process with a large final population $N_{tot}$.

**Modeling the frequency of g-hitchhikers.** We assume a simple and neutral population of cancer cells that grows exponentially with rate $r$. For simplicity, we here assign each new daughter cell one new mutation (alternative mutation rates do not affect the derivation, see Methods). At time $t_1$, a mutation occurs that accelerates the growth rate of the specific subpopulation by a scalar multiplier $k$ such that the new population expands with new rate $k \times r$. At the time of biopsy $T = t_1 + t_2$, where the fitness mutation occurs at $t_1$ and expands for time $t_2$, we expect the frequency of a generational g-hitchhiker mutation that occurred at time $t_m < t_1$ (see Fig. 1 and Methods) to follow a frequency function $f_g$

$$f_g(T, t_m) = \frac{N_R + N_F - N_{RF}}{N_{tot}},$$

or

$$f_g(T, t_m)$$
$$= \frac{e^{-rt_m}\left[N_{tot} - f_{d(T,t_1)} \times N_{tot} + \sqrt[k]{f_{d(T,t_1)} \times N_{tot}}\right] + f_{d(T,t_1)} \times N_{tot} - \sqrt[k]{f_{d(T,t_1)} \times N_{tot}}}{N_{tot}},$$

(1)

where $f_{d(T,t_1)}$ is the frequency of the driver mutation occurring at $t_1$ and expanding for $t_2 = T - t_1$, The terms $\left\{e^{-rt_m} * \left[N_{tot} - f_{d(T,t_1)} * N_{tot}\right]\right\}$ and $\{f_{d(T,t_1)} * N_{tot}\}$ correspond to the growth of regular $N_R$ and fitness $N_F$ populations respectively, while extracting $N_{RF} = \left\{\sqrt[k]{f_{d(T,t_1)} * N_{tot}}\right\}$ for not double counting the hypothetical regular growth of fitness cells.

Eq. (1) for the $m$th hitchhiker implicitly allows one to use the previous $m - 1$ potential hitchhikers to refine the estimates of growth rate $r$ and scalar effect $k$. This estimation is achieved either through a nonlinear-least-squares optimization, and/or through the independent calculation of growth $r$.

The frequency of g-hitchhiking mutations follows the form of an exponential distribution. Theoretically, this further allows us to estimate growth rate $r$ from consecutive g-hitchhiking mutations $m_1$, $m_2$, and $m_3$, which occurred at times $t_{m1}$, $t_{m2}$, and $t_{m3}$ ($t_{m1}$, $t_{m2}$, and $t_{m3} < t_1$), respectively, according to

$$r = \ln\left(\frac{f_g(T, t_{m1}) - f_g(T, t_{m2})}{f_g(T, t_{m2}) - f_g(T, t_{m3})}\right).$$

(2)

In practice, to obtain more accurate estimates, our default algorithm estimates the growth rate $r$ from three more distant time points $t$, $t + n$, and $t + m$ ($n < m$ and $t + m < t_1$) with final frequencies $f_g(T, t)$, $f_g(T, t_n)$, and $f_g(T, t_m)$, respectively (see Methods).

**Optimizing for any time point during tumor progression.** In addition to our independent estimate of growth rate $r$, and in order to avoid previous frequency perturbations in our sample and localize the effect timewise, we also include an extra parameter referred to as "generational time ($t_g$)", which allows us to calibrate an offset for the number of past generations until that point without considering previous mutations outside our sliding window. Thus, similar to Eq. (1), we now have

$$f_g\left(T, t_g, t_i - m\right)$$
$$= \frac{e^{-r(t_g+t_i-m)} \times \left(N_{tot} - f_{d(T,t_i)} \times N_{tot} + \sqrt[k_i]{f_{d(T,t_i)} \times N_{tot}}\right) + f_{d(T,t_i)} \times N_{tot} - \sqrt[k_i]{f_{d(T,t_i)} \times N_{tot}}}{N_{tot}},$$

(3)

where $f_d(T, t_i)$ is the frequency of the putative driver $i$ occurring at time $t_i$.

This approach allows us to reoptimize $t_g$ at any time $t_i$ during tumor growth, independently of earlier or later calculations.

**Birth-and-death model, Gillespie simulations.** First, we tested our algorithm on simulated data based on various growth models, including (a) exponential growth, (b) exponential growth with delayed cell division, and (c) logistic growth (birth-and-death model). We performed simulation models (a) and (c) using a stochastic Gillespie algorithm, whereas model (b) represents an exponential cell growth model with a lag time for cell division, which prevents a cell from redividing immediately. Briefly, for the Birth-and-Death Gillespie model, which is the workhorse of our simulations, we used a stepwise time-branching process to model the growth of a single transformed cell into a tumor with a dominant subclone. At each time step, an event type is chosen with a probability proportional to the event's prevalence. Then, a cell of the eligible type is randomly chosen to undergo that event. In our logistic-growth simulations, the death rate of each cell climbs proportionally as carrying capacity is reached, whereas in our exponential simulations, the death rate of each cell is constant throughout the simulation. The simulation ends randomly, after the driver subclone reaches a critical prevalence. The Gillespie algorithm has been frequently used to simulate stochastically dividing cells[46–52], although simulations with special attention to cell cycle have also been recommended[53].

During simulated growth, we assigned a driver mutation with additional propagating effects from nearly neutral to high ($k = 1.1$, 2, 3, and 4), thus leading to faster growth for the respective subpopulation that contains the specific mutation. Using conservative assumptions, these scalar values represent a range of projected selection coefficients $s^*$ from 0.001 to 0.03 in biologically sized populations (see Methods). For each simulation, we calculated each mutation's frequency in the total population and ordered them based on that frequency. Then, by applying our method we calculated the ranking distance $D$ (as the number of ordered mutations) between the true and our predicted driver (growth peak), as well as the driver's scalar effect $k$.

We tested our method's performance in simulated tumors of lower coverage and different effects. Higher sequencing depth and scalar effect $k$ provided more accurate results and improved our method's implementation (Fig. 2a, b). Lower coverage was associated with worse $k$ calculations and driver predictions, as well as lower positive-predictive values (PPVs). For weak drivers, low sequencing coverage made their identification more difficult. Absolute median ranking distance $|\widetilde{D}|$ was 41 for coverage $100\times$ / $k = 2$, compared to 13 for coverage $1000\times/k = 2$ and $|\widetilde{D}| = 11$ for coverage $1000\times/k = 4$, respectively. In general, driver identification required either a higher than $100\times$ coverage, or a stronger effect (i.e., $k > 2$, $s^* > 0.01$ for a projected cell population of 1,000,000 cells) (Fig. 2i).

Overall, we were able to well approximate the driver's occurrence and effect (Fig. 2). For the birth-and-death model with simulated coverage $1000\times$, the median predicted estimation for simulated effects $k = 2$, $k = 3$, and $k = 4$ was 2.3, 2.9, and 3.8, respectively (Fig. 2ii). Moreover, the median ranking distance $\widetilde{D}$ between simulated and predicted drivers with effect $k = 1.1$ (nearly neutral), $k = 2$, and $k = 3$ was 71, 3. 5, and 6, respectively. The corresponding median distances for random mutations were 73, 43, and 41 (Supplementary Fig. 1). For our nearly neutral simulations ($k = 1.1$, $s^*$ ~0.001 for a projected cell population of 1,000,000 cells) the median distance $\widetilde{D}$ in driver predictions and random predictions was very similar and not significant.

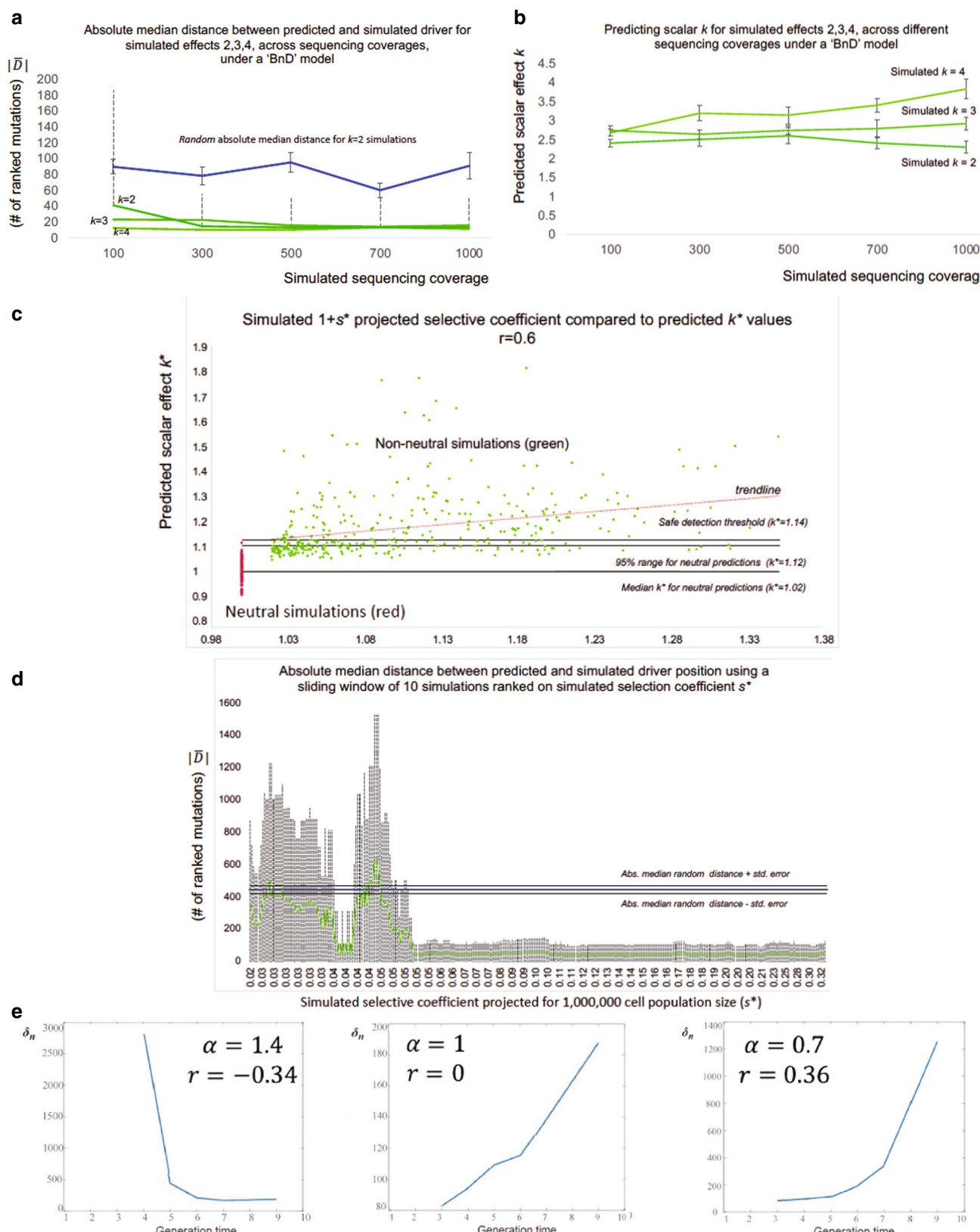

**Simulations with added stochasticity in mutation rates**. To further test our model on a separate independent simulation dataset, we applied our method to (a) neutral simulations of tumor progression and (b) nonneutral simulations for various growth scenarios, as previously developed and described by Williams et al.[16,54] (see also Supplementary Methods). These simulations, although also based on the Gillespie growth model, included added stochasticity with varying mutation rates during tumor progression ($\bar{\mu} = 10$ mutations per cell division). For every simulation, both neutral and nonneutral, we identified our model's highest predicted effect peak, calculated the effect $k$ and absolute median ranking distance $\widetilde{|D|}$ between the simulated and

**Fig. 2 Deeper coverage and stronger drivers improve predictions.** In **a**, using 541 simulations of tumor growth under a birth-and death model, we show the absolute median distance $\widetilde{|D|}$ as in "absolute number of ordered mutations" between predicted and simulated driver for sequencing depths. With the exception of $k = 2$ for 100× (two-tailed $t$ test $P = 0.015$), we were able to detect the driver's presence ($P < 0.005$). Blue line represents the random $\widetilde{|D|}$ as derived by selecting a random mutation from each simulation and calculate the absolute distance to the simulated driver. Dotted lines represent the $2 \times \sigma$ deviation from $\widetilde{|D|}$ while capped bars the median's standard error. For convenience, we only show bars for $k = 2$. In **b**, Using the same simulations, lower coverage provides less accurate $k$ predictions with a lower effect. Capped bars represent the standard error of the median effect prediction. The three lines represent simulations with simulated effect of 2–4. In **c**, using the "Williams et al. 2018" algorithm, we simulated 360 nonneutral and 140 neutral tumors for 10,000 cells. Then, we adjusted our effect predictions for $n^\star$ equal to 1,000,000. In addition, we also adjusted the simulated selection coefficient $s^\star$ for the same populations. Pearson's $r$ between the simulated adjusted coefficient "$1 + s^\star$" against adjusted predicted $k^\star$ was 0.6. In **d**, after ranking $s^\star$ for every nonneutral simulation, we used a sliding window of 20 simulations to estimate $\widetilde{|D|}$ (and $2 \times \sigma$) between the simulated and predicted driver within every window. Dotted lines represent $2 \times \sigma$ deviation. When $s^\star > 0.05$ our driver detection became highly accurate. Blue line represents $\widetilde{|D|}$ for random predictions (444.5), while black lines represent median standard error (24.5). Simulated $s^\star$ have been projected for $n^\star = 1,000,000$. In **e**, using Kingman's coalescent theory, we show that growth estimator $\hat{r}$ remains qualitatively unchanged even for non g-hitchhikers. As mutational density $\delta_n$ increases with $n$, and hence with time, $\hat{r}$ estimator is predicted to take positive values for both constant and varying populations. Similarly, for negative growth, $\delta_n$ decreases with time. We let $\alpha > 1$ corresponding to a decreasing and $\alpha < 1$ corresponding to an increasing population.

predicted driver in number of ranked mutations. Various scenarios for nonneutral growth included a wide range of simulated selection coefficients $s$ (0–33, for a population size of 10,000 cells), categorized driver's VAF (small 0.1–0.2; medium 0.2–0.3; large 0.3–0.4) and larger cell population projections using population genetic models and method adjustments. Corresponding neutral simulations were also generated using the same population parameters. Overall, and in agreement with our previous analyses, our results suggest a small overlap between neutral and nonneutral peaks for weak drivers (Fig. 2c and Supplementary Fig. 1f) and highly significant driver predictability when the predicted driver effect was larger than our (narrow) neutral-effect distribution (Fig. 2c, d and Supplementary Fig. 1g–i). For instance, for simulated populations of 10,000 cell without projection (0 < simulated $s$ < 33) and 1000× coverage our method provided accurate driver detections when the predicted effect was larger than $k = 1.29$ with $\widetilde{|D|}$ ~50 mutations compared to 444.5 for random. These results are directly comparable to our previous analyses, considering the new mutation rates. Similarly, for a projected cell population of 1,000,000 cells, our method provided accurate driver detection for projected selection coefficient $s^\star >$ 0.05 (Fig. 2d). Larger population projections typically decreased the predicted effect $k^\star$ and selection coefficient $s^\star$, but did not affect our method's ability to detect drivers (Supplementary Fig. 1k) as these projections also decreased the standard deviation of our neutral-effect distribution (predicted $k^\star$ for neutral-effect peaks). When we combined 140 neutral with 360 nonneutral simulations, drivers with medium final VAF showed the highest correlation between simulated selection coefficients and our method's predicted scalar $k$ effects (Pearson correlation $r = 0.60$, Fig. 2c). Drivers with lower final VAFs (small ~0.1–0.2) provided slightly lower correlation but had the highest driver detectability, with $\widetilde{|D|} = 46$ mutations between the simulated and predicted driver (Supplementary Fig. 1l), where random $\widetilde{|D|}$ was 444.5 mutations. A (tenfold) higher $\widetilde{|D|}$ here is expected since for these simulations we assumed 10 instead of 1 mutation per cell division.

**Estimator $\hat{r}$ for non g-hitchikers**. We also tested the behavior of the estimator for $r$ (Eq. (2)) on non-g-hitchhiking mutations (i.e., when the assumption that the mutations are generational hitchhikers is not satisfied). For this purpose, we used coalescent theory to estimate the variation in density of mutations across the VAF spectrum for a variety of models (see Methods). We first analyzed the behavior in a constant-size population, and then in populations with increasing and decreasing exponential growth. Our analysis shows that the growth indicator does not

qualitatively change its behavior in this context, so that negative values continue to represent periods of negative growth, and large positive values represent periods of positive growth. However, here we expect a small positive value in the case of zero growth (Fig. 2e, Supplementary Fig. 2).

**Growth patterns in 993 linear tumors from PCAWG**. Using 993 linear tumors from the PCAWG consortium, we explored the different patterns and dynamics of tumor growth based on our model's assigned growth rates. Tumor linearity (where no parental subclone has two or more children subclones) further ensures that tumor subclones do not intermingle and that higher VAF is associated with earlier occurrence. We note that mutational frequency as described in our equations corresponds to 2× VAF, with correction for purity and copy-number variations. These VAF corrections were obtained from PCAWG and are not implemented by our method, which only considers a final mutational frequency. Using our model, each mutation $i$ from sample in our database is assigned a potential positive or negative growth value $r_i$ and a driver effect $k_i$. Under ideal conditions, for each sample, a vector of effect peaks $r_{i-1} \times k_i$ corresponds to potential drivers at position $i$. However, noise, coverage, and growth stochasticity can cause these peaks to represent the potential presence of a nearby driver, especially in low-coverage sequenced tumors (see Fig. 3a, b).

To identify growth patterns across individual tumors, we (i) normalized each mutation's growth rate based on the sample's maximum growth value; (ii) divided the ordered mutations into 20 bins; and (iii) applied $k$-means clustering to the average normalized value per bin. Our results highlighted three main clustering patterns (Fig. 3c). As expected, most tumors ($n = 525$) showed logistic growth with an increasingly higher growth rate at the beginning and a stabilization at the later stages. For many tumors ($n = 366$), an early high growth period was followed by a stagnation and potential reduction in tumor size. This effect could also be artificially enhanced due to sampling errors for mutations with low VAF (during late tumor progression). The last group of tumors ($n = 102$) showed relatively steady, continuous growth. However, it is uncertain whether this pattern represents tumors that were sequenced early. Further, some types of cancer seemed to prefer specific growth patterns (Fig. 3c).

By modeling tumor growth, we can find mutations during positive or negative growth periods in single or multiple individual samples. Through positive growth enrichment, we characterized the degree to which one type of mutation (e.g., TSGs/*TP53*, nonsynonymous) or region (e.g., *TP53*) was significantly enriched and associated with periods of positive growth across multiple samples. We then compared each

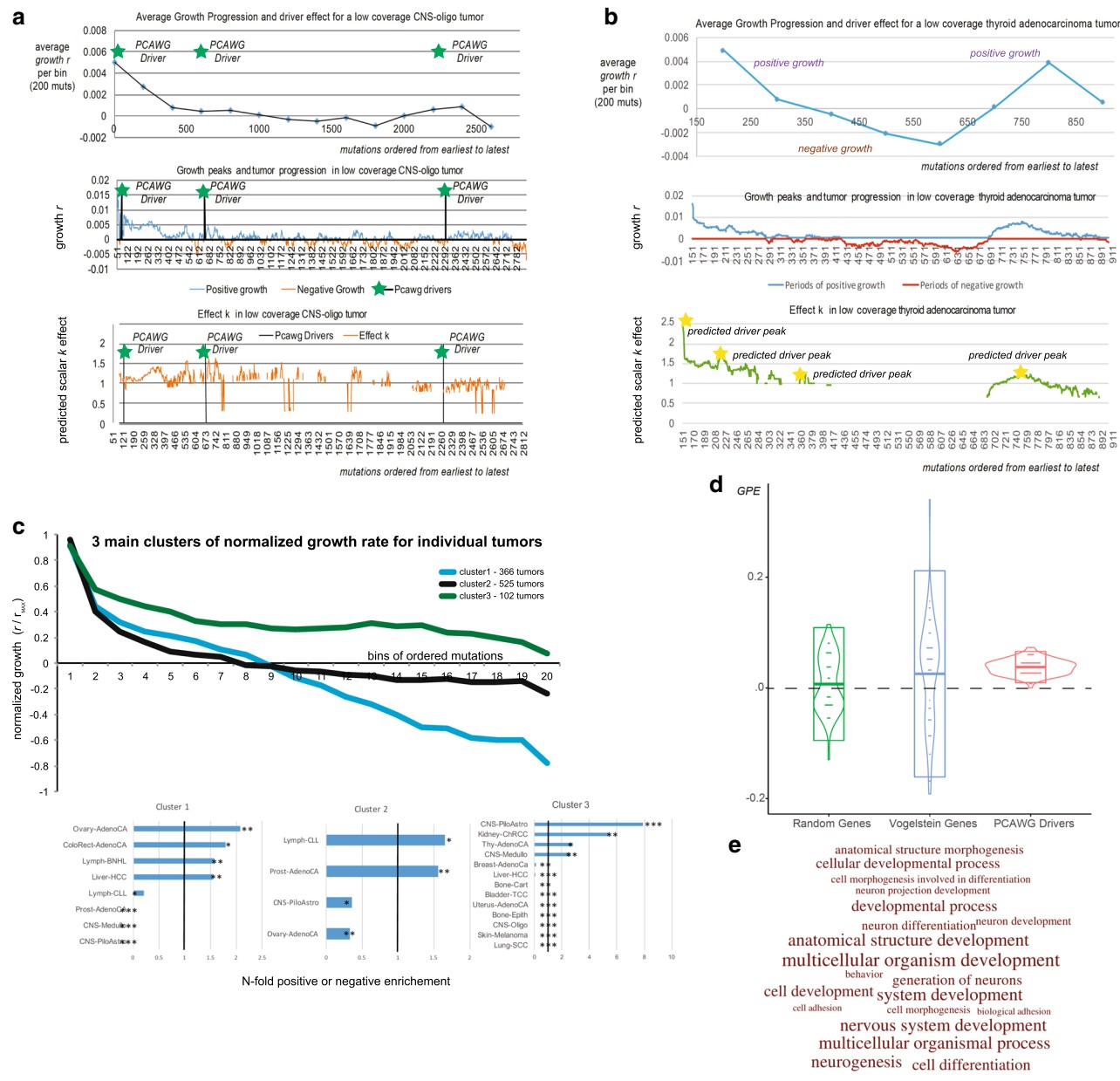

**Fig. 3 Growth patterns and growth associations using 993 linear tumors.** Across 993 linear tumors from PCAWG consortium we expect an under-selection mutation to be associated with periods of positive growth. We compared several mutation types (driver mutation, mutation within geneX, within GO categoryX), to a random distribution from their respective sample for association with positive growth. **a**, **b** The averaged growth progression, mutational growth, and mutational effect, for a single low-coverage CNS-oligo tumor and a single low-coverage thyroid adenocarcinoma tumor without any PCAWG-identified drivers. Green asterisks denote the ordered position of a PCAWG-predicted driver within the sample. Yellow asterisks denote a growth peak and putative driver presence. In **c**, we derived three main growth patterns (steady growth, sigmoid growth, and stagnation/shrinkage) for 993 linear tumors, as they were grouped using a k-means clustering algorithm. Various cancer types showed specific enrichment or depletion for the three clusters (levels of significance for Fisher's tests for enrichment noted as *, **, and *** for p < 0.05, 0.01, and 0.001). In **d**, PCAWG drivers and Vogelstein genes show significant positive growth enrichment compared to a list of random highly mutated genes. Boxplots represent 2 × σ deviation, lines represent the mean, while violin plots are trimmed to data range. **e** We show the GO enrichment for the 20 most affected biological processes, when we use 293 genes, significantly associated with periods of positive growth.

mutation type to random mutations from their respective samples. To confirm whether we could detect any signal of selection at the gene level, we compared positive growth enrichment (PGE) for mutations between (i) the Vogelstein gene list[28]; (ii) a comparable list (in mutational numbers) of randomly selected genes; and (iii) a list of assigned drivers from the PCAWG consortium[9,32]. As expected, PCAWG-assigned driver SNVs clearly showed the highest positive enrichment, followed by SNVs that were not individually called by PCAWG as drivers but

that fall within the Vogelstein driver gene list (Fig. 3d). We note, however, that our random gene list did show a small positive enrichment, as this list contains several often-mutated genes and potential drivers or mini-drivers. We obtained similar results when we repeated the comparison while considering the difference between additional mutational effect against a random distribution (Supplementary Fig. 3).

In an effort to better understand the microenvironment of tumor dynamics, the selective forces, and the biological processes

that are most keenly affected by tumor progression, we analyzed a list of 1000 most mutated genes in the PCAWG samples where we identified 293 genes with significant overall association with positive growth (Supplementary Data 1). Then we further tested these genes for Gene Ontology (GO) enrichment. As expected, developmental and differentiation processes were highly enriched during periods of positive growth, showing signals for being under positive selection. Interestingly, we found that genes related to multicellular processes showed the highest enrichment based on raw $p$ value (Fig. 3e, Supplementary Data 2).

**TSGs vs. oncogenes.** Based on each mutation's genomic properties (e.g., genomic position, coding vs. noncoding, TSG vs. oncogene, cancer type, and GO annotation), we can examine whether the specific type of mutation (or mutation element) is statistically enriched during periods of positive growth when compared to random mutations from their respective samples. However, the more specifically that we defined a mutation type, the fewer mutations that corresponded to this category. For example, the Vogelstein TSGs in our dataset contain 321 missense and 103 nonsense mutations, whereas *TP53* in our dataset contains 71 nonsynonymous mutations and 13 nonsense mutations. Unfortunately, for many tumor genes and cancer types, we currently have a small number of mutations, precluding significance in the results.

A recent study by Kumar et al. suggested that high-impact mutations should have more clear positive effects on tumor growth when they are located in TSGs vs. oncogenes[37]. This is expected, as generally a "defected" oncogene with reduced expression should not favor cancer progression. To better understand the behavior of TSGs and oncogenes, we tested for positive enrichment of synonymous, nonsynonymous, premature stop, promoter, and intronic mutations (Fig. 4). As expected, our results showed significant enrichment of missense and nonsense mutations in TSG regions. During periods of positive growth, 45 nonsense and 128 missense mutations corresponded to an average of 37.4 and 117.96 random mutations, respectively (100 bootstraps replicates, $p$ values = 7.823348e−30 and 1.632649e −23). Interestingly, promoter and intronic regions also showed a significant positive effect on tumor growth, suggesting that some noncoding mutations in TSGs might favor positive growth (Fig. 4a).

In the case of oncogenes, we did not find significant enrichment of missense mutations, but we did find significant association between their promoter regions and positive growth (Fig. 4b). This might be due to many reasons including the pancancer nature of our analysis, lack of power and small sample size, our modeling assumptions, or the noise due to low sequencing coverage per tumor sample. However, many genes including oncogenes might be under negative selection, with only a small subset of their respective mutations being favorable to cancer growth. Moreover, high-impact mutations in oncogenic regions do not necessarily favor tumor growth. Indeed, our data contain only four nonsense mutations in oncogenic regions. Some oncogenes such as *MET* and *CTNNB1* showed slight overall negative enrichment, but their nonsynonymous mutations, especially in specific cancers, showed enrichment during periods of positive growth (Supplementary Fig. 3).

To detect mutations during positive growth periods, we applied our model to individual types of mutations (i.e., missense, synonymous, intronic, nonsense, and promoter) for each Vogelstein gene. Overall, our results identified various mutation elements including promoters, nonsense, and missense with significant effects (Fig. 4c). Interestingly, synonymous *BLC2* mutations that occurred near an early positioned mutational

hotspot were significantly associated with positive growth (Fig. 4c and Supplementary Fig. 4). Synonymous mutations are not generally considered to be important in cancer; however, previous studies have reported recurrent synonymous F17F mutations in *BLC2*-like 12, where regulatory hsa-miR-671-5p alters the gene's expression[44].

**Growth peaks and driver effects on a model AML tumor.** In addition to the 993 PCAWG low-coverage tumor samples, we implemented our model on an ultra-deeply sequenced AML (>250×) liquid tumor. A ultra-deeply sequenced tumor provides more accurate global VAF, which should in turn allow for better estimation of model parameters[12].

In general, the predicted peaks of our model mapped very closely to mutations from known cancer genes (Fig. 5). Deep valleys followed by the highest growth peaks corresponded with close approximation to the three missense mutations from known cancer genes (*IDH1*, *IDH2*, and *FLT3*, $p$ value < 2.2e−16). Thus, in agreement with previous studies[34,35], the derived growth patterns suggested three to five major genetic hits from cancer mutations in order to render tumor growth permanent.

In addition, we used all the mutations in our previous database to evaluate those in the deeply sequenced AML in order to identify new candidates associated with positive growth. As a result, we further identified five additional candidates from the ultra-deep AML sample that belong to genomic elements associated with positive growth (Fig. 5d). These additional candidates consist of four missense mutations (*SRCAP*, *CPS1*, *GLI1*, and *COL18A1*) and one intronic mutation (*MAP3K1*), which appeared to align near observed, previously unexplained periods of initial growth. Previous recent studies have also linked *CPS1* and *GLI1* to various cancers[55–58]. Finally, based on our PCAWG database, for each driver candidate we detected possible positive enrichment across varying effect ranges [0.9, 1.1, 1.3, 1.5, 1.7, 1.9, and 2.1] (Supplementary Fig. 5). Indicatively, our independent estimation of mutational effect suggested a high correlation when compared to the calculated effect using the deep-sequenced model AML tumor (Supplementary Fig. 5).

## Discussion

Most approaches to identify driver candidates are based on recurrent mutations and large cohorts[22]. More recently, studies have probed tumor selection either through deviation from background metrics or by using VAF distribution to quantify the subclonal effect[20]. Here, we present a framework that models tumor progression using generational hitchhikers and localized time reoptimizations using mutational frequencies from individual samples to (i) determine periods of positive or negative growth, (ii) suggest the presence of candidate drivers and estimate their effect on tumor progression, and (iii) detect genomic regions or mutation elements that are associated with positive or negative growth periods. Overall, our work highlights the importance of whole genome deep sequencing for modeling tumor progression.

When we applied our framework to 993 individual tumors from the PCAWG consortium, our growth analysis indicated different growth patterns across cancer types, including steady growth, sigmoidal growth, and modes of stagnation. Determining tumor progression can be useful in understanding each tumor's historic aggressiveness, and the effect of driver mutations on tumor progression (VAFs used by our method typically represent past growth, as latest mutations tend to have undetected frequency in our sample). In addition, we identified several biological processes significantly affected by tumor progression, including genes involved in multicellularity. These results might

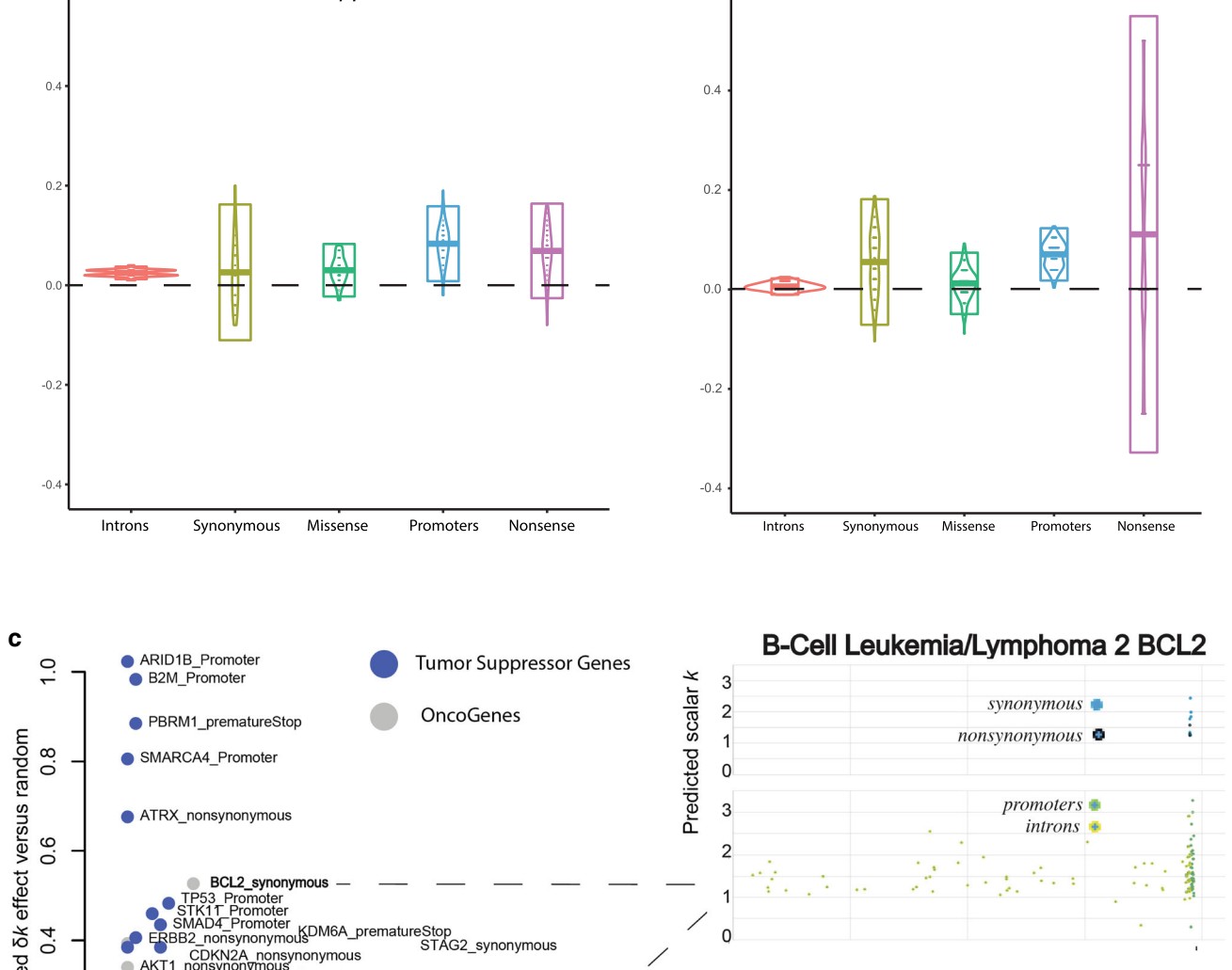

**Fig. 4 Tumor-suppressor gene and oncogene elements show growth enrichment.** We show the positive growth enrichment across different mutation types (introns, synonymous, missense, nonsense, and promoters). For **a**, Vogelstein tumor suppressor genes and **b** Vogelstein oncogenes, boxplots represent $2 \times \sigma$ deviation, lines represent the mean, while violin plots are trimmed to data range. In **c**, we plot gene elements (e.g., {GeneX_mutation type}) from Vogelstein gene list that showed significant positive or negative enrichment. We further zoom in to *BCL2*'s genomic region to map missense, nonsynonymous, promoter, and intronic mutations.

indicate an evolutionary transition during tumor progression from multi-cell functionality to single-cell selection.

As expected, we found significant enrichment of known PCAWG drivers, Vogelstein cancer genes, and nonsense and missense mutation TSGs during periods of positive growth. In accordance with some previous studies[40–43], our results also suggested that a small proportion of intronic mutations could affect TSGs (but not oncogenes), whereas some synonymous mutations could affect oncogene (but not TSG) expression. Even though defective splicing in TSGs or changes in the negative

regulation of oncogenes are not entirely unexpected[44], noncoding mutations are not generally considered to be major driver events in tumor progression. Thus, it is possible that our results are subject to analytical (e.g., model parametrization, initial parameters, window size selection, low sequencing coverage, and sample size) and biological (e.g., hitchhiking) error.

Using variant-allele frequency to quantify driver effects and tumor progression can be challenging. Our analysis might be subject to different types of bias, including sequencing noise, growth stochasticity, model parameterization, low sequencing

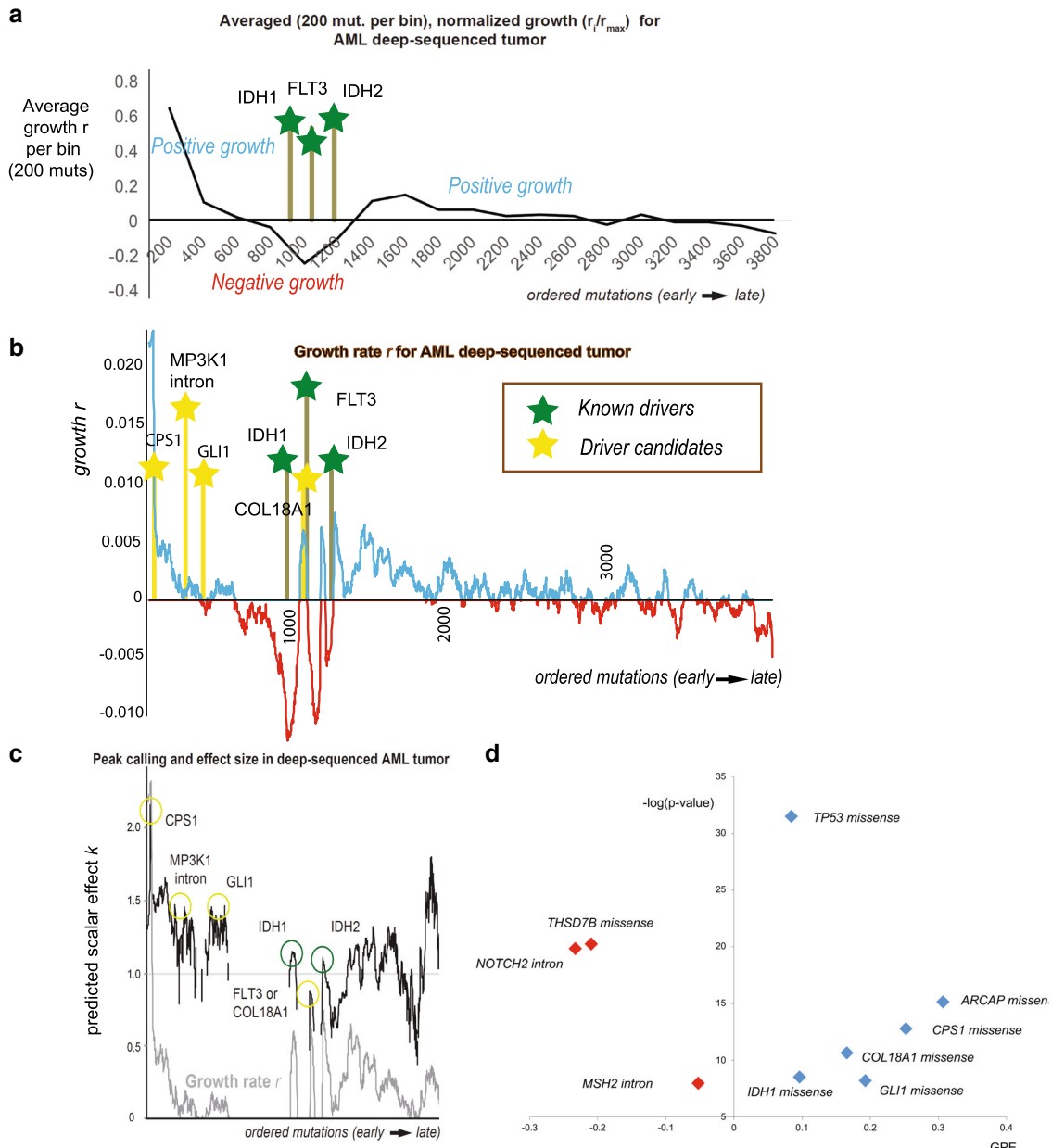

**Fig. 5 Tumor progression on model AML liquid tumor.** In **a**, we show the averaged growth progression for an AML ultra-deep-sequenced tumor. We ordered the sample's mutations from highest to lowest frequency and divided them into bins of 200 mutations. Three cancer mutations hit the tumor to establish a permanent growth (cancer mutations denoted by green bars). In **b**, we plot the mutational growth $r_{i-1}$ for each mutation across tumor progression. The three cancer genes (*IDH1* missense, *FLT3* missense, and *IDH2* missense) aligned well with 3 of our top 5 growth peaks (two-tailed $t$ test $p < 2.2e-16$). Candidate driver mutations -denoted by yellow bar—that we identified from our PCAWG database as being associated with positive growth (see also "(d)") aligned well with early—previously unjustified growth peaks. In **c**, we show each mutation's effect in tumor progression. Effect peaks corresponds to putative drivers. **d** By using our PCAWG database from our previous analysis, we tested which mutations from the deep-sequenced sample were associated with positive growth. The $x$-axis represents positive growth enrichment, while the $y$-axis shows the level of significance as the negative logarithm of a two-tailed $t$ test $p$ value ($-\log(p \text{ value}) > 5$). Overall, we found 6 mutation types that showed significant positive enrichment across 993 PCAWG tumors, including *TP53* missense (appeared during metastasis), *IDH1* missense, *COL18A1* missense, *CPS1* missense, *GLI1* missense, and *SRCAP* missense. Missense *TP53* and *SRCAP* mutations are not included in graph (b) as they were metastatic mutations. For association with positive growth we tested all missense mutations (e.g., *CPS1* missense), and every mutation in the sample from Vogelstein cancer genes (e.g., *NOTCH2* intron).

coverage, tumor ploidy, subclonality, and a low number of tumor samples per cancer or mutational element. Under a neutral model, our method would still detect some growth peaks or suggest the presence of weak drivers. These are false-positive predictions, possibly due to noise which results in various signal perturbations in the VAF spectrum, or potential genetic drift. Moreover, our model does not consider the potential effects from deleterious passenger mutations or sequencing errors on the VAF spectrum. However, we consider that -if not depleted- most deleterious mutations should have a small VAF in our sequenced sample. Similarly, we expect that sequencing errors tend to produce spurious mutations of extremely low VAF, which are ignored by our framework. Although some researchers are skeptical of the plausibility of VAF quantification[19,59], recent

analyses have also confirmed that it can be achieved even at low sequencing coverage[16]. At the same time, as sequencing cost decreases exponentially, ultra-deep whole-genome sequencing for a larger number of samples will become trivially within reach. This is critical for the personalized assessment and parametrization of single samples.

Similar to previous Darwinian, bacterial, and viral evolution analyses, modeling the variations of cell populations allows us to associate these variations with specific events, even at a single sample level. Our work contributes to our understanding of cancer evolution by directly assessing tumor sample progression at the time of the driver event. This assessment can be very critical for therapeutic strategies and drug selection[32,33]. Our framework presents opportunities for personalized diagnosis via modeling the tumor's progression using deep-sequenced whole-genome data from one single individual.

## Methods

**Modeling the frequency of g-hitchhiking mutations**. Let us assume that a simple population of cancer cells grows exponentially; for simplicity, we assign one mutation per cell division.

$$N(t) = e^{rt}. \tag{4}$$

At sequencing time $T$, the frequency of a mutation occurring at time $t_n$ would be equal to

$$f_n(T, t_n) = \frac{e^{r(T-t_n)}}{e^{rT}} = e^{-rt_n}. \tag{5}$$

At time $t_1$, a mutation occurs that increases the growth rate $r$ of the specific subpopulation by $k$, such that the new population is now expanding as

$$N_F = e^{krt}. \tag{6}$$

Thus, at total time $T = t_1 + t_2$, we expect a generational $(g-)$ hitchhiking mutation that occurred at time $t_m < t_1$ (see Fig. 1) to have a frequency equal to

$$f_g(T, t_m) = \frac{e^{r(T-t_m)} + N_F - e^{rt_2}}{N_{tot}}, \tag{7}$$

were $N_{tot}$ is the total number of cells (or mutations) and $N_F$ is the number of cells that contain the fitness mutation that occurred at $t_1$ and expanded for $t_2$.

Thus, $N_F = e^{krt_2}$. $\tag{8}$

or

$$e^{rt2} = \sqrt[k]{N_F},$$

and Eq. (7) can be re-written as

$$f_g(T, t_m) = \frac{e^{r(T-t_m)} + N_F - \sqrt[k]{N_F}}{N_{tot}}. \tag{9}$$

Moreover, if we assume that $t_m \sim t_1$

then $f_g(T, t_m \sim t_1) = \frac{e^{r(t_1+t_2-t_m)} + N_F - \sqrt[k]{N_F}}{N_{tot}} = \frac{e^{rt_2} + N_F - \sqrt[k]{N_F}}{N_{tot}} = \frac{N_F}{N_{tot}}$, or

$$\lim_{t_m \sim t_1} f_g(T, t_m) = \frac{N_F}{N_{tot}} = f_d(T, t_1). \tag{10}$$

Eq. (9) can be rewritten as

$$f_g(T, t_m) = \frac{e^{r(t_1+t_2-t_m)} + N_F - \sqrt[k]{N_F}}{N_{tot}} \Rightarrow$$

$$f_g(T, t_m) = \frac{e^{r(t_1+t_2)} \times e^{-rt_m} + N_F - \sqrt[k]{N_F}}{N_{tot}} \Rightarrow$$

$$f_g(T, t_m) = \frac{(N_{tot} - N_F + e^{rt_2}) \times e^{-rt_m} + N_F - \sqrt[k]{N_F}}{N_{tot}} \Rightarrow \tag{11}$$

$$f_g(T, t_m) = \frac{(N_{tot} - N_F + \sqrt[k]{N_F}) \times e^{-rt_m} + N_F - \sqrt[k]{N_F}}{N_{tot}}$$

which is the frequency function $f_g$ for the g-hitchhiking mutations in our sample.

Finally, according to (10) we get

$$f_g(T, t_m)$$
$$= \frac{e^{-rt_m} \times \left[ N_{tot} - f_{d(T,t_1)} \times N_{tot} + \sqrt[k]{f_{d(T,t_1)} \times N_{tot}} \right] + f_{d(T,t_1)} \times N_{tot} - \sqrt[k]{f_{d(T,t_1)} \times N_{tot}}}{N_{tot}}, \tag{1}$$

We note that the frequency $f_g(T, t_m)$ of g-hitchhiking mutations also follow the form of an exponential function.

$$f_g(T, t_m) = A \times e^{-rtm} + B. $$

This allows the sampling and estimation of growth rate $r$ from consecutive g-hitchhiking mutations $m_1$, $m_2$, and $m_3$ that happened at corresponding times $t_{m1}$,

$t_{m2}$, and $t_{m3}$ according to

$$r = \ln \left( \frac{f_g(T, t_{m1}) - f_g(T, t_{m2})}{f_g(T, t_{m2}) - f_g(T, t_{m3})} \right). \tag{2}$$

**Independent calculation of growth $r$ for g-hitchhikers**. For three hitchhiking mutations that occured at times $t$, $t + n$, and $t + m$ ($n < m$), their respective frequencies are $f(t)$, $f(t + n)$, and $f(t + m)$

$$\text{Let } \Lambda = \frac{f_g(T, t) - f_g(T, t + n)}{f(T, t) - f_g(T, t + m)} = \frac{(1 - e^{-rn})}{(1 - e^{-rm})}. \tag{12}$$

Thus,

$$\Lambda \times e^{-r \times m} - e^{-r \times n} - \Lambda + 1 = 0.$$

If we set $e^{-r} = x$
then

$$\Lambda \times x^m - x^n - \Lambda + 1 = 0.$$

By selecting $m = 2 \times n$

$$\Lambda \times x^{2n} - x^n - \Lambda + 1 = 0.$$

Therefore

$$x^n = \frac{1 \pm \sqrt{1 - 4 * \Lambda (-\Lambda + 1)}}{2 * \Lambda} = e^{-r \times n}$$
$$x = e^{-r} = \sqrt[n]{\frac{1 \pm \sqrt{1 - 4 * \Lambda(-\Lambda+1)}}{2 * \Lambda}} \tag{13}$$
$$r = -\log \left( \sqrt[n]{\frac{1 \pm \sqrt{1 - 4 * \Lambda(-\Lambda+1)}}{2 * \Lambda}} \right)$$

**Optimizing for $t_g$ at any time point during progression**. In (1), we associated g-hitchhiker frequency with population growth $r$ as

$$f_g(T, t_m) = \frac{e^{-rt_m} \times \left[ N_{tot} - f_{d(T,t_1)} \times N_{tot} + \sqrt[k]{f_{d(T,t_1)} \times N_{tot}} \right] + f_{d(T,t_1)} \times N_{tot} - \sqrt[k]{f_{d(T,t_1)} \times N_{tot}}}{N_{tot}}.$$

Furthermore, we included an extra parameter for "generational time" ($t_g$) allowing us to optimize for the number of generations until that point without knowledge of previous mutations.

Thus

$$f_g\left(T, t_g, t_i - m\right)$$
$$= \frac{e^{-r(t_g + t_i - m)} \times \left( N_{tot} - f_{d(T,t_i)} \times N_{tot} + \sqrt[k_i]{f_{d(T,t_i)} \times N_{tot}} \right) + f_{d(T,t_i)} \times N_{tot} - \sqrt[k_i]{f_{d(T,t_i)} \times N_{tot}}}{N_{tot}}, \tag{3}$$

where $f_d(T, t_i)$ is the frequency of the putative driver $i$ occurring at time $t_i$.

This allows us to reoptimiIe $t_g$ at any time $t_i$ during tumor growth independent of earlier calculations.

This allows us to

1. *exclude* pretumor somatic mutations or duplications in our calculations,
2. reoptimize at any time $t$ during tumor growth independent of earlier calculations.

**Behavior of $\hat{r}$ estimators on nongenerational mutations**. We used the coalescent theory to analyze the behavior of the estimator

$$\hat{r} = \log \left( \frac{f_g(T, t) - f_g(T, t + n)}{f_g(T, t + n) - f_g(T, t + m)} \right), \tag{14}$$

when the assumption that the mutations are generational is not satisfied. Our results indicate that the growth indicator does not change qualitatively. We first analyzed the behavior in a constant-size population, and then in populations with increasing and decreasing exponential growth.

**Behavior of $\hat{r}$ estimators for constant population sizes**. We consider a population of constant size $N$ with mutation rate $\mu$. Given the population observed at a fixed time point $t_0$, we can consider the coalescent tree of all cells at $t = 0$ reaching back to their most recent common ancestor at $t = T$ (where time is indexed in reverse direction from $t_0$). Writing $T_n$ for the length of time over which $n$ lineages are present (i.e., the time between the splits of lineage $n$ and $n + 1$, hence $T = \sum_{n=1}^{N} T_n$, where $T_N$ is truncated at time 0), using Kingman's coalescent[60] it can be shown that

$$T_n = \frac{2}{n(n-1)}. \tag{15}$$

Assuming that the birth-and-death rates remain constant, the number of mutations $M_n$ acquired during $T_n$ can be expressed up to a constant of proportionality

$$M_n \propto \mu n T_n = \frac{2\mu}{n-1}. \quad (16)$$

We can approximate the variation in density of the VAF spectrum by assuming that all mutations falling in $T_n$ take their expected frequency $f = 1/n$, and calculating the density $\delta_n$ within windows $[1/n1/(n-1))$, whose lengths are $L_n = 1/(n-1) - 1/n = 1/(n(n-1))$

$$\delta_n = \frac{M_n}{L_n} \propto 2\mu n. \quad (17)$$

As $\delta_n$ increases with $n$, and hence with time, $\hat{r}$ as estimated using Eq. (14) is predicted to take positive values in a constant-size population.

**Populations with varying size**. To predict the behavior of $\hat{r}$ in populations of varying size, we formulated the coalescent as a Markov process. We let $t = 0$ represent the time of observation and indexing time in reverse as above, and $X_t$ for a random variable represent the number of lineages in the coalescent tree at time $t$, and $N_t$ for the population size at time $t$. To define a Markov process over $X_t$ from $t = 0...T$, we fixed the initial distribution to $p(x_0 = i) = [i = N]$, where $[.]$ is the Iverson bracket. We also defined a transition matrix $\tau$ such that $\tau(i,j)$ represents the conditional probability $p(x_{t+1} = j | x_t = i)$; that is, the probability that there are $j$ coalescent lineages at time $t + 1$ given there are $i$ lineages at time $t$. Note that the number of coalescent lineages present will be less than or equal to the size of the population at a given time.

To calculate $\tau(i, j)$, we evaluated the number of maps that take $i$ lineages to $j$ lineages given the final population size of $N_{t+1}$. As there are $j$ final lineages, the image of the map must have size $j$, meaning that it must be one of the $\binom{N_{t+1}}{j}$ subsets of the population at $t + 1$. For each of these subsets, the original $i$ lineages can be partitioned into those taking distinct values at $t + 1$, so that there are $I$ possible partitions of $i$ lineages into $j$ nonempty subsets, where $\begin{Bmatrix} i \\ j \end{Bmatrix} = \frac{1}{j!}\sum_{k=0}^{j}(-1)^{j-k}\binom{j}{k}k^n$ is a Stirling number of the second kind. Further, there are $(j!)$ permutations of the image set to which each of these partitions caI be mapped. Given $N_{t+1}^i$ maps in total from the populations from $t$ to $t + 1$ when restricted to the ancestors in the coalescent tree

$$\tau(i,j) = \frac{\binom{N_{t+1}}{j}\begin{Bmatrix} i \\ j \end{Bmatrix}(j!)}{N_{t+1}^i} = \binom{N_{t+1}}{j}\frac{\sum_{k=0}^{j}(-1)^{j-k}\binom{j}{k}k^n}{N_{tI+1}^i}I, \quad (18)$$

assuming $j \leq i$, and $\tau(i, j) = 0$ otherwise. To investigate the behavior of $\hat{r}$, we used the Markov chain above to calculate $p_{t+1} = p_t\tau$ for a fixed number of time-steps, where $p_t = [p(x_t = 1), p(x_t = 2), ..., p(x_t = N)]$. We then calculated a function $g(t)$ representing the expected number of lineages present at time $t$, $g(t) = \sum_n np(x_t = n)$. We calculated $T_n$, which estimates the length of time over which there are exactly $n$ lineages as above as

$$T_n = \min(\{t | g(t) \leq n\}) - \min(\{t | g(t) \leq n-1\}), \quad (19)$$

from which the total number of mutations $M_n$ acquired during $T_n$ can be calculated using Eq. (16), and the variation in density of the VAF spectrum over windows corresponding to the intervals $T_n$ can be calculated using Eq. (17).

We calculated $\delta_n$ as a function of $n$ in a number of populations, using the population model $N^{t+1} = \alpha N^t$, where we let $\alpha = [1,1.1,1.2,...,2]$, corresponding to a decreasing population (as time is indexed in reverse), or $\alpha = [1,0.9,0.8,...,0.5]$, corresponding to an increasing population. We used 200 time-steps for all calculations with $\mu = 0.01$. We started all decreasing populations at $N_0 = 10$ and fixed a maximum population size of $N_{max} = N_0\alpha^{10}$, while fixing a minimum size for all increasing populations at $N_{min} = 10$, and starting at $N_0 = N_{min}\alpha^{-10}$. For all $t$ after the population reaches its maximum/minimum size, we set $N^{t+1} = N^t$. Supplementary Fig. 2 shows the output for populations of decreasing and increasing sizes, respectively. As predicted by the earlier analysis, the calculations show that $\delta_n$ is an increasing function for constant population size ($\alpha = 1$), corresponding to a positive value of $\hat{r}$, and is approximately linear. Likewise, $\delta_n$ is increasing for all populations of increasing size (Supplementary Fig. 2b); hence, $\hat{r}$ is predicted to be positive for all such populations, with a magnitude increasing with $\alpha$, as the rate of increase of $\delta_n$ increases for larger $\alpha$. For decreasing populations (Supplementary Fig. 2a), $\delta_n$ is only strictly decreasing for $\alpha > 1.4$, corresponding to $r = -\log(\alpha) \approx -0.34$ in generational units, suggesting that a negative $r$ that is at least this magnitude will result in a negative estimate for $\hat{r}$ using Eq. (14).

**Reconsidering the assumption of one mutation per division**. In our model, we have assumed for reasons of convenience and simplicity that one new mutation arises per cell division. However, this assumption is not required to implement our model. To derive the estimator for $r$ in Eq. (2), all that is required is that the intervals $t_{m2} - t_{m1}$ and $t_{m3} - t_{m2}$ are equal in expectation. For a mutation rate 0.5 \\$\mu = 1$ (where \\$\mu$ is the total number of mutations expected per cell division), this interval is one generation, but for \\$\mu < 2$ the expected interval is $2/\backslash\mu$.

**Model optimization and initial parameters**. To optimize our model, we used custom perl scripts and the R package "Nonlinear Least Squares" (NLS)[61,62] with sliding windows of $m = 150$ g-hitchhikers. We optimized for [["mod $< -\text{nls}(\overrightarrow{f} \sim \exp(-r \ast (t_g + \overrightarrow{mut\_order}))\ast(1 - \alpha) + \alpha$, start = list($\alpha = 0.01$, $t_g = 1$), control = nls.control(maxiter = 10,000,000, tol = 1e-04, minFactor = 0.000002, printEval = TRUE, warnOnly = TRUE))"]], where $f$ is the frequency vector for the g-hitchhikers, $t_g$ corresponds to generational time, and $\alpha$ is a composite parameter associated with the driver's prevalence and its respective effect according to equation

$$f_g\left(T, t_g, t_i - m\right)$$
$$= \frac{e^{-r(t_g+t_i-m)} \times \left(N_{tot} - f_{d(T,t_i)} \times N_{tot} + \sqrt[k]{f_{d(T,t_i)} \times N_{tot}}\right) + f_{d(T,t_i)} \times N_{tot} - \sqrt[k]{f_{d(T,t_i)} \times N_{tot}}}{N_{tot}},$$

where $f_d(T, t_i)$ is the frequency of the putative driver $i$ occurring at time $t_i$.

Growth $r$ can be estimated using equation

$$r = \ln\left(\frac{f_g(T, t_{m1}) - f_g(T, t_{m2})}{f_g(T, t_{m2}) - f_g(T, t_{m3})}\right),$$

or alternatively through NLS optimization.

When the population shows an exponential growth and we are only interested in the first driver or the subclonal effect, we can omit the generational time ($t_g$) estimation to reduce unnecessary optimizing errors. In this case, we simplify the R command

[["mod $< -$ nls($\overrightarrow{f} \sim \exp(-r \ast \overrightarrow{mut\_order})\ast(1 - \alpha) + \alpha$, start = list($\alpha = 0.01$), control = nls.control(maxiter = 10000000, tol = 1e-04, minFactor = 0.000002, printEval = TRUE, warnOnly = TRUE))"]] or [["mod $< -$ nls($\overrightarrow{f} \sim \exp(-r \ast \overrightarrow{mut\_order}) \times \alpha + \beta$, start = list($\alpha = 0.01$, $\beta = 0.1$), control = nls.control(maxiter = 10000000, tol = 1e-04, minFactor = 0.000002, printEval = TRUE, warnOnly = TRUE))"]].

**Finding the best sliding window size m**. For our PCAWG analysis, we used a sliding window of $m = 100$ and 150 g-hitchhikers. Our presented results were based on $m = 150$. Overall, larger windows provided a more stable uniform analysis across all 993 tumors, allowing the NLS algorithm to converge easier across all samples, by considering a larger range of mutation frequencies in tumors with lower coverage. However, especially in deep-sequenced tumors, the size of a sliding window should be individually optimized based on population assumptions. For our simulation analyses, we selected the size of a sliding window that minimizes the absolute median ranking distance $||\widetilde{D}|$ between true and predicted drivers by maximizing the $p$ value when compared to a distance distribution of 100 random mutations. For our independent set of nonneutral simulations based on Williams et al. software, we were able to calculate an optimal window size by tuning our algorithm based on 464 nonneutral simulations. The optimal window size that provided a median effect of 1 for the neutral simulations was 150 hitchhikers, which is also what we used for the deep-sequenced AML tumor. Smaller window sizes provided a higher median effect for both neutral and nonneutral simulations, without burdening our method's detectability.

**Scalar k and selection coefficient s**. In real tumors, cells bearing a subclonal driver mutation can form a distinguishable subclone within a tumor of millions, billions or trillions of cells, as a result of small growth advantage of these cells compounded over hundreds to thousands of generations. The $k$ values used in our simulations should therefore be scaled when predicting the corresponding $k\ast$ values in a real population on which our estimators would exhibit similar behavior (due to similar amounts of variance/genetic drift). Felsenstein[60] describes scaling rules for simulations: to use a smaller population to simulate a larger one, the quantity $4 \times N \times s$, where $N$ is the population size and s the selection coefficient must remain the same. We consider a range of population sizes ($10^6$–$10^{10}$) as being realistic (Williams et al.[16] use an estimate of $10^{10}$ cells; we note however that spatial effects may result in a lower effective population size in many tumors). Using the scaling $k\ast = 1 + \frac{N_s(k-1)}{N_r}$, where $N_s$ and $N_r$ are the simulation and realistic population sizes respectively, the range of $k$ we considered in our simulations from 1.1 to 4, corresponds under this scaling to $k\ast \in [1.001, 1.03]$, which are noticeably smaller than 1.1 ($s\ast = 0.1$ corresponding to a very strong driver effect). We consider these values to be upper-bounds, as the true effective population size is likely to be substantially larger than one million cells for most cancers. As an alternative to scaling the values of $k$ as discussed, we also consider the effects of directly substituting realistic size estimates ($10^6$ to $10^{10}$) into the variable $N_{tot}$ in equation $f_g\left(T, t_g, t_i - m\right) =$
$$\frac{e^{-r(t_g+t_i-m)} \times \left(N_{tot} - f_{d(T,t_i)} \times N_{tot} + \sqrt[k]{f_{d(T,t_i)} \times N_{tot}}\right) + f_{d(T,t_i)} \times N_{tot} - \sqrt[k]{f_{d(T,t_i)} \times N_{tot}}}{N_{tot}}.$$ As shown in Supplementary Fig. 1, this leads to an improvement in the accuracy with which we detect simulated drivers (in terms of the distance from the simulated driver). Our simulations thus imply that our algorithm can detect drivers with weak effects accurately in tumors of realistic sizes.

**Simulating tumors of lower coverage depth in sequencing**. To simulate sequencing coverage depth of 100×, 300×, 500×, and 700×, we first created a cell population from 1000× coverage based on mutational frequencies. In this sense, a mutation with a frequency of 0.475 would be associated with 475 out of 1000 individuals that contained the specific mutation (noted as "1") and 525 individuals that did not (noted as "0"). Consequently, we sampled with replacement sub-populations with sizes 100, 300, 500, and 700 cells. Then we recalculated the lower coverage frequencies as the sum of "1"s divided by the corresponding coverage.

**True positives, false positives, sensitivity, and PPV analysis**. To test our model's performance across simulations with various driver effects and depth coverage we aimed to determine the number of true positive (TP), false positive (FP), true negatives (TN) and false negative (FN) predictions. Based on our previous calculations, we estimated the absolute median distance ($|\widetilde{D}|$) between the true and the predicted driver's position close to 11 mutations and the standard error (SE) about 2.5 mutations. For every simulation, a driver prediction was considered as TP, if $|\widetilde{D}|$ between our predicted driver and the true driver was less than $|\widetilde{D}| + 2\text{SE} = 16$ mutations. If the predicted driver peak was at distance longer than 15 mutations, the corresponding predicted driver was considered as FP. If a simulation provided zero TPs, we then considered the lack of TP as FN. Similarly, if a simulation resulted in zero FPs, we then considered the lack of FP as a TN. We should note that the cut off of 15 mutations is fairly strict for the detection of TPs, especially for a real size tumor/samples, but helpful to systematically evaluate our model across different simulations.

**Ranking distance D (between true driver and effect peak)**. For each simulation, we calculate the ranking distance $D$ (as a number of ordered mutations) between the ranked position of simulated driver and the predicted growth or effect peak (predicted driver). We further calculate the median $\widetilde{D}$ and absolute median distance $|\widetilde{D}|$ for different $k$ effects, across simulated sequencing coverage, population size, etc. To assess statistical significance for our method's ability to detect drivers, for each simulation, we also draw a random driver prediction. Then, for each sample we calculate the distance between our random prediction and the simulated driver as previously described. Finally, we compare the median, absolute median, standard deviation, and SE against the random distance distribution for all simulations. $p$ Values are obtained from a two-tailed $t$-test. In samples where the true driver is unknown (e.g., PCAWG linear tumors), the ranking distance $D$ from a growth peak corresponds to the ranked number of order mutations for one specific mutation from the closest peak (e.g., PCAWG drivers).

**Positive growth enrichment**. A type of mutation (e.g., *TP53* missense) is assessed if it occurs significantly more often than random during periods of positive growth $r$. For every type of mutation that we tested, we picked an equal number of random mutations from the same individual samples. By repeating this process 100 times we set our mean expectation for randomly associating mutations with positive growth. Then we assess if the specific type of mutation is enriched during periods of positive growth compared to random. For example, for $n = 48$ "*TP53* missense" mutations in our sample with positive growth, we found a comparative average of $\overline{x_{\text{pr}}} = 40.38$ of random mutations with a SE of mean $\text{SEM} = 0.42$. Significance was then assigned based on $z$-score. As PGE we report the value of $\text{PGE} = \frac{(x_p - \overline{x_{\text{pr}}})}{\text{total}\#(\text{of e.g. TP53 missense})}$, where in this example, $x_p$ is the number of *TP53* missense mutations found during positive growth and $\overline{x_{\text{pr}}}$ is the average number of random mutations with positive growth $r$, as sampled 100 times with replacement.

**Significance for deep-seq AML tumor drivers**. To test the level of significance for our growth peak prediction in the deep-sequenced tumor, we selected our top five highest growth peaks and estimated the distance $D$ between the three known cancer genes and the closest growth peak. Then, for 1000 replicates we sampled random mutations with replacement to create a random distribution of distances between a random mutation and its closest peak. For a more conservative approach, we increased the number of highest peaks to ten and reduced the random mutation sample to the first 2000 mutations without losing significance. A two-tailed $t$ test was performed to establish the level of significance.

**Enrichment across effect bins**. To estimate enrichment across different effect ranges for specific mutation types (e.g., *TP53* missense mutations) we created effect bins of $k = [0.9–1.1, 1.1–1.3, \ldots 2.1–2.3, 2.3–2.5, 2.5–2.7]$. Across the PCAWG samples, for each mutation, we also picked one random mutation from the same sample. Then, we bootstrapped this process for 100 replicates. Finally, for every bin we tested whether the specific mutation appeared to be enriched compared to random.

**Reporting summary**. Further information on research design is available in the Nature Research Reporting Summary linked to this article.

## Data availability

PCAWG protected datasets are controlled access that is subject to data usage agreement. PCAWG datasets are available upon request and authorization from the ICGC Data Access Compliance Office and dbGaP Authorized Access program for US-based projects. For data repositories and data request see https://docs.icgc.org/pcawg/data/. Pseudo-VCF files are provided at https://doi.org/10.6084/m9.figshare.9722651.v1. These files contain real VAF distributions, mutation type information including gene names, but all genomic coordinates and variance information have been masked and randomly modified. The source data underlying Figs. 2–5 and Supplementary Figs are provided as a Source Data file.

## Code availability

We are providing a perl script that analyzes a pseudo-VCF derived file format for growth and effect calculation. It should be noted that our model does not correct for purity and ploidy inconsistencies, but instead utilizes already derived mutational frequencies. Our code is publicly available, together with test play data and a readme file at https://github.com/gersteinlab/Evotum101.

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

## Acknowledgements

We thank the PCAWG consortium for the state-of-the-art preliminary analysis and management of the data. We thank the members of the Gerstein lab Cancer Genomics group (S.K., S.L., and P.D.M.) for helpful discussion on the method development and analysis of the data. We acknowledge support from the NIH Grant (R01 HG 008126). This work was partially supported by NIH/NIGMS T32 GM007205.

## Author contributions

L.S. conceived of the project, designed, performed, and analyzed the experiments. W.M. designed and developed the simulations. J.W. performed the coalescent analysis and designed the simulations. L.S. drafted the paper. L.S. and M.G. wrote the paper. All authors read and approved the final paper.

## Competing interests

The authors declare no competing interests.
