## [Peer Review File · Nature Communications]

Reviewers'

comments:

Reviewer #1 (Remarks to the Author):

Salichos and colleagues developed a new method to pinpoint the fitness effects of individual point mutations on tumor progression. Based on a deterministic model of tumor progression with the assumption that every cancer cell has exactly one surviving descendant with one additional mutation, they show that the observed frequency of these mutations in an exponentially growing population can be exploited to find individual driver mutations from many hitchhiking mutations.

I very much like the idea of the paper to identify individual drivers as it circumvents the problem of identifying the actual driver mutations even when the driver gene is known for this particular cancer type. Perhaps the authors want to discuss in their introduction that a large fraction of mutations in driver genes are not true drivers of tumor progression which is also clinically a critical problem for drug selection (Reiter et al, Science 2018, Tamborero et al, Genome Medicine 2018). Therefore, new methods such as the one proposed in this study are important.

Comments and questions:

1. For better understanding it would be helpful to state early in the manuscript that this is a deterministic model to avoid any confusion since many statements do not hold in a stochastic model of growth and mutation appearance.
2. Was the full mutational frequency spectrum considered for the analysis or did the authors focus on windows of VAFs such as Williams et al (Nature Genetics 2016) on 12%-25%? I assume that similarly the method struggles with ordering mutations with an almost clonal or almost absent frequency.
3. More details of the simulations are necessary to understand the model validation (starting page 11). Since the calculations were based on a deterministic model, what was the exact simulated birth-death process? There is an enormous body of work on stochastic models of tumor progression. Did the authors reuse any established and validated model?
4. What happens if a neutral model of tumor progression is simulated (Sottoriva et al, Nature Genetics 2015)? Are any driver mutations identified?
5. What are the observed false-positive and false-negative rates for various driver mutation rates and fitness effects?
6. How were the parameter values chosen? A paragraph on parameter selection would be generally helpful. For example, the parameter fitness effect k (equivalent to $(1+s)$ where s is the selection parameter in the cancer population genetics literature) has been estimated to be $s=10\%$ ($k=1.1$) for very strong drivers. Nevertheless, the authors explored k values of 2, 3, and 4 and described $k=1.1$ as nearly neutral. What were the used division and death rates? Was the population sequenced at a specific size or time?
7. How did the initial growth rate affect the results? How did the death rate affect the results? How did the passenger mutation rate affect the results? Did the authors assume a sequencing error model? If yes, which one, with which error rate?
8. How was the mutation appearance simulated in the non g-hitchhikers?
9. Figures: I was unable to read many labels in the figures because they were too small. E.g., Fig. 2: the tick labels appear at font size 1 while the x axis label might be 10 and the y axis label and the legend might be 20.
10. Figures: Some panels are missing x-axis or y-axis labels. E.g., Fig. 3c
11. Page 12, line 247: Hitchhikers spelling
12. Page 14, line 295: Wrong reference for a PCAWG consortium driver list. Reference 9 does not provide a PCAWG driver list.
13. Page 27, line 585: remove redundant "and"
14. Supplementary table 1. Should be converted into a computer-readable format.
15. Code was not available for review and is also not provided in a repository.

Overall, I think that this study has a lot of potential. Nevertheless, the clarity and the presentation could be improved. I mostly struggled with some missing intermediate steps in various sections, the readability of the figures and the insufficient benchmarking to fully validate the results. More benchmarking across many different scenarios of tumor progression and realistic parameter values would strongly improve this study.

Reviewer #2 (Remarks to the Author):

The manuscript describes a framework for identifying mutated driver genes based on Variant Allele Frequencies in individual deeply sequenced whole genome sequencing data. In contrast, existing methods tend to use the count of recurrently mutated genes. This is potentially a very important and useful new approach.

The manuscript is far from publishable. The writing makes the manuscript very difficult to read. The text needs additional polishing to make it more easily understood. The derivation in the supplementary material is somewhat easier to follow. I'd suggest an early reference to this in the main manuscript. However, contains several errors (e.g. line 182, hitchhiker (sic), line 317 uniformal (sic)).

The methodology seems generally sound, though this is difficult to establish completely due to the quality of the writing and in the absence of the code. The results recapitulate results on "test" data, but this is currently also unpublished. Some justification is needed of why more extensive benchmarking is not required.

It might make more sense to largely remove the mathematical description from the main part of the manuscript, and replace it with a higher level description, while further polishing the Supp Material.

In summary, I'd need to see a more polished version of the manuscript before I could declare the manuscript publishable.

Specific comments:

The code is only available on request. This is unacceptable for reviewers or for subsequent sharing if the manuscript is published. Code and test data should be made available to reviewers. As this is a methods paper, the documentation and code need to be seen to establish that they are appropriate and the code even works.

What about CNVs? How do you deal with their impact on VAFs? It is mentioned that they are ignored (p6, line 131). The supplementary material also superficially mentions ploidy and purity (line 7). Are CNV regions filtered out, as some other methods do? If this is the case, that is potentially fine as a starting point (as has been done previously in clonality estimation and other areas), but it needs to be made clear. This is simply not dealt with adequately. The lack of available code also makes this impossible to understand or assess.

How deeply does sequencing need to be performed? How is accuracy impacted as depth is decreased?

The model assumes that there are no deleterious mutations. What would be the impact on the model and results if both advantageous and deleterious mutations occurred. This should be mentioned.

The use of quotes in defining mathematical symbols (e.g. "k", 'k', "effect k", "r",) is problematic and causes inconsistency. I think these symbols should be defined clearly as rates (r), or scalar multipliers (k), or as appropriate. This is dealt with somewhat better in the Supp Material.

The term linear tumours should be defined the first time it is used. It is actually the expansion or evolution that is linear, so this needs to be clarified the first time the term appears, as well as in the abstract.

The term total time should be avoided.

Lines 144-155 (especially line 150) are completely impenetrable, and should be polished and clarified.

Line 145: "...our framework models the function f_g that provides the prevalence...". Prevalence is a poor choice of word that makes this unclear. I'd also say the framework uses the model function f_g , rather than models the function.

Supp material, line 93: sequencing time would be better termed time of biopsy or sample collection.

Lines should not start with commas (e.g. lines 175, 181, etc).

Line 243 & 245: Define median distance clearly.

We would like to thank the reviewers for their positive and constructive comments which we would like to address one -by-one below, for reasons of convenience.

Amongst many changes in our revised manuscript, as per our reviewers' suggestions:

- We now offer a public demo version of our code together with test-data that we have created and provide at <https://github.com/gersteinlab/Evotum101>.
- We have extensively revised our manuscript for clarity and comprehension. Moreover, we have moved most of the mathematical equations into supplement which we have replaced with a higher-level description of our model.
- We have performed additional analyses to test our method's behavior. These analyses are best depicted in our renewed 'figure 2' of our manuscript. Previous figure 2 has been moved to supplement as figure S6.

1.1

Referee Comment	R1.1: Salichos and colleagues developed a new method to pinpoint the fitness effects of individual point mutations on tumor progression. Based on a deterministic model of tumor progression with the assumption that every cancer cell has exactly one surviving descendant with one additional mutation, they show that the observed frequency of these mutations in an exponentially growing population can be exploited to find individual driver mutations from many hitchhiking mutations. I very much like the idea of the paper to identify individual drivers as it circumvents the problem of identifying the actual driver mutations even when the driver gene is known for this particular cancer type
Author Response	We thank the referee for the positive feedback.
Excerpt From Revised Manuscript	NA

1.2

Referee Comment	R1.2: Perhaps the authors want to discuss in their introduction that a large fraction of mutations in driver genes are not true drivers of tumor progression which is also clinically a critical problem for drug selection (Reiter et al, Science 2018, Tamborero et al, Genome Medicine 2018). Therefore, new methods such as the one proposed in this study are important.
-----------------	---

Author Response	Indeed, part of the motivation of our work is the clinical importance of variants of unknown significance. The revised manuscript discusses these variants and cites the suggested articles.
Excerpt From Revised Manuscript	Manuscript highlighted lines: 94-96, 456-457 “More importantly... drug selection ^{33,34} ” “This assessment can be very critical for therapeutic strategies and drug selection.”

1.3

Referee Comment	For better understanding it would be helpful to state early in the manuscript that this is a deterministic model to avoid any confusion since many statements do not hold in a stochastic model of growth and mutation appearance.
Author Response	In the revised manuscript, we clarify upfront that we assume a deterministic model allowing for stochastic growth. We derived our estimators for r , k , and t_g through analyzing a deterministic model, which is the limit in expectation of a stochastic exponential model for a large N . Our analyses of simulations using the Gillespie algorithms show that these estimators can be applied to a range of stochastic models to yield realistic parameter estimates.
Excerpt From Revised Manuscript	Manuscript highlighted lines: 164-166 “ We derived our estimators ... large final population N_{tot}. ”

1.4

Referee Comment	Was the full mutational frequency spectrum considered for the analysis or did the authors focus on windows of VAFs such as Williams et al (Nature Genetics 2016) on 12%-25%? I assume that similarly the method struggles with ordering mutations with an almost clonal or almost absent frequency.
Author Response	When working with empirical data, our analysis considered mutations across the VAF spectrum. When working with simulations, our standard analysis used a VAF cutoff of 0.05. Williams et al. focused on neutral evolution. In neutral evolution, the VAF order of subclonal mutations is best distinguished within the 12-25% VAF window. In contrast, our focus is on tumors in which a subclone

	may have a positive fitness advantage. In such tumors, hitchhiking mutations of the dominant subclone will have a VAF distribution compressed around the VAF of the subclonal driver, which is frequently but not always in the 12-25% window. Thus, in the tumors we are considering, the 12-25% VAF window does not have special significance. Moreover, our method does not rely exclusively on the order of individual mutations but on the VAF discrepancies in sliding windows of mutations, which are more robust to noise. We describe the supporting technical details behind this approach in a section of the revised supplement entitled “Independent calculation of growth r for g-hitchhikers”
Excerpt From Revised Manuscript	Manuscript highlighted lines: 193-201 “Theoretically, this further allows.. described in the Supplement.” Supplement lines: 184-216 “This allows the sampling... [s14]”

1.5

Referee Comment	More details of the simulations are necessary to understand the model validation (starting page 11). Since the calculations were based on a deterministic model, what was the exact simulated birth-death process? There is an enormous body of work on stochastic models of tumor progression. Did the authors reuse any established and validated model?
Author Response	We used a stepwise time-branching process to model the growth of a single transformed cell into a tumor with a dominant subclone. The workhorse of our simulations is the Gillespie algorithm, which researchers have frequently used to simulate stochastically dividing cells (see Baar et al. 2016; Castellanos-Moreno et al. 2014; Ryser et al. 2016; Figueredo et al. 2014; Turner et al. 2009; Mort et al. 2016; Zaider and Minerbo 2000). At each time step, an event type (such as reproduction or death of a clonal or subclonal cell) is chosen with a probability proportional to the total rates of those possible events, where the total rate of an event is the sum of the event rate across all cells. Then, a cell of the eligible type is randomly chosen to undergo that event. The birth rate of cells is proportional to their fitness, which increases in cells that have inherited a subclonal driver mutation. The death rate of all cells is assumed to be homogenous at a given time step. In our logistic-growth simulations, the death rate of each cell climbs proportionally as carrying capacity is reached, whereas in our exponential simulations, the death rate of each cell is constant throughout the simulation. In a standard run of our simulations, each cell division produces a new mutation, whose ancestry is recorded. When there is no subclonal driver extant in the tumor, each new mutation has a fixed chance of being a subclonal driver. The simulation ends a short, random time after the driver subclone reaches a critical

	prevalence. More details and parameter values are listed in the updated sections in both manuscript and supplement.
Excerpt From Revised Manuscript	Manuscript highlighted lines: 226-237 “Briefly, for the “ Birth and Death ” Gillespie model... have also been recommended ⁵⁵ .” Supplement highlighted lines: 378-424 “ Simulation analysis using the Gillespie algorithm ... critical prevalence. ”

1.6

Referee Comment	What happens if a neutral model of tumor progression is simulated (Sottoriva et al, Nature Genetics 2015)? Are any driver mutations identified?
Author Response	Under a neutral model, our method would still detect some growth peaks or suggest the presence of weak drivers. These are false positive predictions, possibly due to noise which results in various signal perturbations in the VAF spectrum, or potential genetic drifts. In figure S2, we show that under a nearly neutral model of tumor progression, with a scalar multiplier $k=1.1$ which corresponds to a very weak population-scaled s of 1.001, we were not able to significantly distinguish true drivers from random peaks, (i.e. very weak drivers could not create subclones overcoming drift from random mutations or noise artifacts). As shown in our new figure 2 in the manuscript, our method’s positive predictive values [PPV=TP/(TP+FP)] improve as the sequencing depth and/or driver effect get higher. In our simulations, to detect true driver signal we need at least 100x coverage or $k > 2$ ($s \sim 0.01$). We have now included a phrase in the discussion about the identification of putative drivers in the presence of noise or genetic drift.
Excerpt From Revised Manuscript	Manuscript highlighted lines: 248-255, 262-264, 439-451 “ We tested our method’s... stronger effect (i.e. $k>2$) (Figure 2). ” “ For our nearly neutral simulations ($k = 1.1, s \sim 0.001$) the median distance \tilde{D} in driver predictions and random predictions was very similar and not significant. ” “ Under a neutral model... parametrization of single samples. ” Supplement: 459-471 “ True positives, false positives,... model across different simulations. ”

1.7

Referee Comment	What are the observed false-positive and false-negative rates for various driver mutation rates and fitness effects?
Author Response	We have added a new figure (figure 2) that shows our method's positive predictive value based on the number of false positives and false negatives for various effects and coverage. We did not assess the false-positive and false-negative rates as a function of driver mutation rates, because we restricted the driver mutation rate to be one enduring subclonal driver per tumor for all tumors for simplicity.
Excerpt From Revised Manuscript	Manuscript highlighted lines: 248-255 “We tested our method's performance... stronger effect (i.e. $k > 2$) (Figure 2).” Supplement highlighted lines: 450-471 “Simulating tumors of lower... model across different simulations.””

1.8

Referee Comment	How were the parameter values chosen? A paragraph on parameter selection would be generally helpful. For example, the parameter fitness effect k (equivalent to $(1+s)$ where s is the selection parameter in the cancer population genetics literature) has been estimated to be $s=10\%$ ($k=1.1$) for very strong drivers. Nevertheless, the authors explored k values of 2, 3, and 4 and described $k=1.1$ as nearly neutral. What were the used division and death rates? Was the population sequenced at a specific size or time?
-----------------	---

Author
Response

The design paradigm for our simulations was to reproduce the essential subclonal structure of tumors as simply as possible to test whether we could infer subclonal fitness effects under a set of assumptions. We experimented with parameter values to find ones that reproduced this subclonal structure. Our simulations involved *nominal* k values from 1.1 to 4. However, as it was not possible to simulate biologically accurate tumor sample sizes (which can contain millions of cells), and because the effective selection coefficient of a mutation depends on population size, we conservatively estimated our nominal k values to represent effective selection coefficients s between 0.001 and 0.03 when scaled to the population sizes of real tumors. The division rate of transformed cells without a subclonal driver was on average one cell division per (arbitrary) unit time. The death rate of cells in the logistical model was proportional to the tumor population at a given time, divided by the carrying capacity. The population was sequenced at a short random interval after the subclonal driver reached a prevalence value drawn from a range of acceptable prevalence values. The updated manuscript and supplement explains these model choices in more detail.

Excerpt From Revised Manuscript	Manuscript highlighted lines: 241-243 “Using conservative assumptions, these scalar values represent a range of selection coefficients s from 0.001 to 0.03 in biologically sized populations (see Supplement).” Supplement: 426-448 “Scalar k and selection coefficient s... one million cells for most cancers.”
--	---

1.9

Referee Comment	How did the initial growth rate affect the results? How did the death rate affect the results? How did the passenger mutation rate affect the results? Did the authors assume a sequencing error model? If yes, which one, with which error rate?
Author Response	In the simulations, time is measured in units relative to the initial growth rate. Therefore, the initial net growth rate is arbitrary and cannot affect simulation behavior. Both increasing the death rate and increasing the passenger mutation rate leads to a higher effective number of mutations per successful cell division. Theoretically, doubling the effective number of mutations per successful cell division – either through an increased death rate or an increased passenger mutation rate – will double our estimate of the growth rate r. The value of r is actually a growth rate of mutations rather than of cells, but should not affect our estimate of a scalar multiplier. Practically, a larger number of mutations allows us to better average the VAF noise and test smaller VAF windows. We assumed that there were no sequencing errors. Sequencing errors tend to produce spurious mutations of extremely low VAF, which are ignored by our

	framework. In our manuscript we now clarify that our model does not consider sequencing error.
Excerpt From Revised Manuscript	Manuscript highlighted lines: 442-446 “Moreover, our model does not take into account the potential effects from deleterious passenger mutations or sequencing error on the VAF spectrum.”

1.10

Referee Comment	How was the mutation appearance simulated in the non g-hitchhikers?
Author Response	Mutation appearance was simulated identically in all cells. In standard runs of our simulation, as time marched forward, each new daughter cell acquired one new mutation. At the end, each mutation has a population frequency whether it is a g-hitchhiker or not. We note that, although we assumed for convenience that one new mutation per daughter cell arises per cell division in the derivation given in the main text, this assumption is not required. To derive the estimator for r in equation [s10], all that is required is that the intervals $t_{m2}-t_{m1}$ and $t_{m3}-t_{m2}$ are equal in expectation. For a mutation rate $0.5\mu=1$ (where μ is the total number of mutations expected per cell division), this interval is 1 generation, but for $\mu<2$ the expected interval is $2/\mu$.
Excerpt From Revised Manuscript	Supplement highlighted lines:332-338 “ Reconsidering the assumption of one new mutation per cell division In our model, we have assumed for reasons of convenience and simplicity that one new mutation arises per cell division. However, this assumption is not required to implement our model. To derive the estimator for r in equation [s10], all that is required is that the intervals $t_{m2}-t_{m1}$ and $t_{m3}-t_{m2}$ are equal in expectation. For a mutation rate $0.5\mu=1$ (where μ is the total number of mutations expected per cell division), this interval is one generation, but for $\mu<2$ the expected interval is $2/\mu$.”

1.11 – 1.16: typographic and formatting suggestions have been implanted as recommended

1.17

Referee Comment	Code was not available for review and is also not provided in a repository.
-----------------	---

Author Response	We now provide a perl script that summarizes our algorithm's functionality. Within the script, we also include comments to make it more understandable to the reader The scripts will be made available through github.
Excerpt From Revised Manuscript	Supplement highlighted lines: 483-493 “Code and Data Availability... data request see https://docs.icgc.org/pcawg/data/ .”

1.18

Referee Comment	Overall, I think that this study has a lot of potential. Nevertheless, the clarity and the presentation could be improved. I mostly struggled with some missing intermediate steps in various sections, the readability of the figures and the insufficient benchmarking to fully validate the results. More benchmarking across many different scenarios of tumor progression and realistic parameter values would strongly improve this study.
Author Response	We thank the reviewer for these comments. As suggested, we have moved our model's mathematical description in the supplement and tried to improve the readability of the manuscript by including a higher-level description of our model. Moreover, we have included additional simulation analyses in order to evaluate our method 's sensitivity across sequence depth coverage and mutational effects (new figure 2). Finally, we have included a new section that uses population scaling to project our parameter values into realistic size populations. We hope our revised manuscript is easier to read and comprehend for every reader.
Excerpt From Revised Manuscript	Manuscript highlighted lines: 148-201, 239-255 “ Our framework's equations... as described in the Supplement.” “During simulated growth... or a stronger effect (i.e. $k > 2$) (Figure 2). ” New main figure 2 Supplement highlighted lines: 426-471 “ Scalar k and selection coefficient s ... across different simulations. ”

2.1

Referee Comment	The manuscript describes a framework for identifying mutated driver genes based on Variant Allele Frequencies in individual deeply sequenced whole
-----------------	--

	genome sequencing data. In contrast, existing methods tend to use the count of recurrently mutated genes. This is potentially a very important and useful new approach.
Author Response	We thank the reviewer for these comments about the potential significance of our study.
Excerpt From Revised Manuscript	NA

2.2

Referee Comment	The manuscript is far from publishable. The writing makes the manuscript very difficult to to read. The text needs additional polishing to make it more easily understood. The derivation in the supplementary material is somewhat easier to follow. I'd suggest an early reference to this in the main manuscript. However, contains several errors (e.g. line 182, hitchhicker (sic), line 317 uniformal
Author Response	We thank the reviewer for the suggestions. We have revised the manuscript for clarity and removed the mathematical description from our main manuscript according to the referee's later suggestion. (see comment 2.4). Finally, we have have improved our higher-level description of the model, while including an early reference directing the reader from the main manuscript to the supplement.
Excerpt From Revised Manuscript	Manuscript highlighted lines: 129-215 "Method Overview: Clock-like Hitchhikers ...earlier or later calculations."

2.3

Referee Comment	The methodology seems generally sound, though this is difficult to establish completely due to the quality of the writing and in the absence of the code. The results recapitulate results on "test" data, but this is currently also unpublished. Some justification is needed of why more extensive benchmarking is not required.
-----------------	---

Author Response	We cannot currently provide real PCAWG data. Our data, including driver annotation in cancer samples, are controlled access under protection and this manuscript is under an embargo deadline. After 25 th of July 2019 or the publication of the main PACWG marker paper within the next few months, one can apply to the international cancer genome consortium (ICGC) in order to obtain full access to the controlled data (see supplement). However, we agree with the referee's comment and we now provide a perl script combined with a sample of play data (pseudo VCF files) derived from original variant allele frequencies. In the pseudo VCF files that we created, all names/IDs have been removed and the coordinates of point mutations have been randomly altered. The VAFs, however, have remained the same. In this way, the reviewers can better test our method's functionality.
Excerpt From Revised Manuscript	Supplement highlighted lines: 483-493 " Code and Data Availability ... see https://docs.icgc.org/pcawg/data/ ."

2.4

Referee Comment	It might make more sense to largely remove the mathematical description from the main part of the manuscript, and replace it with a higher level description, while further polishing the Supp Material.
Author Response	In accordance with the reviewer's suggestion, we have moved most of the mathematical descriptions from the main part of the manuscript to the newly polished supplement. In the main text, we now include only high-level descriptions and the final model equations. See our response to 2.12 as an example.
Excerpt From Revised Manuscript	NA

2.5

Referee Comment	The code is only available on request. This is unacceptable for reviewers or for subsequent sharing if the manuscript is published. Code and test data should be made available to reviewers. As this is a methods paper, the documentation and code need to be seen to establish that they are appropriate and the code even
-----------------	---

	works
Author Response	The resubmission includes a perl script and test play data derived from true samples. See R2.3.
Excerpt From Revised Manuscript	Supplement highlighted lines: 483-493 “Code and Data Availability... see https://docs.icgc.org/pcawg/data/ .”

2.6

Referee Comment	What about CNVs? How do you deal with their impact on VAFs? It is mentioned that they are ignored (p6, line 131). The supplementary material also superficially mentions ploidy and purity (line 7). Are CNV regions filtered out, as some other methods do? If this is the case, that is potentially fine as a starting point (as has been done previously in clonality estimation and other areas), but it needs to be made clear. This is simply not dealt with adequately. The lack of available code also makes this impossible to understand or assess.
Author Response	Our framework assumes that the frequency of mutations has already been adjusted for purity and ploidy. Importantly, the PCAWG data used have been pre-adjusted for purity and ploidy. We have now revised the Supplement to make this more clear.
Excerpt From Revised Manuscript	Manuscript highlighted lines:286-288 “These VAF corrections were obtained from PCAWG and are not implemented in any way by our method, which only considers a final mutational frequency.”

2.7

Referee Comment	How deeply does sequencing need to be performed? How is accuracy impacted as depth is decreased?
-----------------	--

Author Response	In our revised manuscript, we have performed and provide new analyses to address this comment. In these analyses we evaluated our method by simulating various sequencing depths, from 100x to 1000x coverage, also for different values of scalar k. Overall, higher sequencing depth and higher k provide more accurate results and seem to improve our method's implementation. Lower depth is generally associated with worse k calculations and driver predictions (as in 'distance between predicted and true driver'), as well as lower positive predictive values (PPVs). For weak drivers (e.g. $k=2$), sequencing depth of 100x makes their identification much harder, possibly due to genetic drift, harder parametrization or other artifacts. In general, driver identification requires either a higher than 100x coverage, or for this depth a stronger driver (i.e. $k>2$). In our revised manuscript, we now present this additional analysis combined with a new supplementary figure.
Excerpt From Revised Manuscript	Manuscript lines: 248-264 “We tested our method's performance in simulated tumors of lower coverage and different effects. Higher sequencing depth and scalar effect k provided more accurate results and improved our method's implementation (Figure 2). Lower coverage was associated with worse k calculations and driver predictions, as well as lower positive predictive values (PPVs). For weak drivers, low sequencing coverage made their identification more difficult. Absolute median ranking distance \tilde{D} was 41 for coverage $100x/k=2$, compared to 13 for coverage $1000x/k=2$ and $\tilde{D} =11$ for coverage $1000x/k=4$ respectively. In general, driver identification required either a higher than 100x coverage, or a stronger effect (i.e. $k>2$) (Figure 2).”

2.8

Referee Comment	The model assumes that there are no deleterious mutations. What would be the impact on the model and results if both advantageous and deleterious mutations occurred. This should be mentioned.
Author Response	If deleterious mutations are equally likely to fixate throughout the tumor, they will effectively lower the baseline growth rate r of the tumor without necessarily affecting the fitness impact of the subclonal driver mutation k. We do agree with the reviewer that deleterious mutations might overall burden a subclone and their study might be of great interest. However, it should be mentioned that the study of cancer drivers by the PCAWG consortium did not find many evidence for the excessive presence of deleterious mutations (Rheinbay et al 2017; Sabarinathan et al 2017). In the revised discussion, we acknowledge that we have not explicitly considered deleterious mutations.

Excerpt From Revised Manuscript	Manuscript highlighted lines:442-445 “Moreover, our model does not take into account the potential effects from deleterious passenger mutations or sequencing error on the VAF spectrum. However, we consider that -if not depleted- most deleterious mutations should have a small VAF in our sequenced sample.”
--

2.9

Referee Comment	The use of quotes in defining mathematical symbols (e.g. "k", 'k', "effect k", "r",) is problematic and causes inconsistency. I think these symbols should be defined clearly as rates (r), or scalar multipliers (k), or as appropriate. This is dealt with somewhat better in the Supp Material.
Author Response	We now refer to symbols using italics and clarify that r is a rate (divisions / unit time) and k is a scalar multiplier.
Excerpt From Revised Manuscript	NA

2.10 -2.11; 2.13-2.16 – terminology and spelling have been updated as suggested

2.12

Referee Comment	Lines 144-155 (especially line 150) are completely impenetrable, and should be polished and clarified.
Author Response	We have now revised these lines and replaced this text with a high-level summary, and moved the technical details to the supplement.

Excerpt From Revised Manuscript	Manuscript lines: 129-166 “ Method Overview: Clock-like Hitchhikers, Growth Rates, Local Re-optimization, and Driver Effects ... with a large final population N_{tot}. ”
--	---

Reviewers' comments:

Reviewer #1 (Remarks to the Author):

The authors have partially addressed my questions. Nevertheless, the benchmarking should be significantly extended. Figure 2 is a first step but this is far away from a comprehensive evaluation what a reader would expect for a methods paper in Nature Communications. There are no error bars. How many runs were simulated? No statistical tests. Values for all parameters should be provided when essential for understanding a figure. Because of the rather unusual model setup, there is a lot of uncertainty around the parameter values (in particular k), hence a much large range of values should be explored. With this very minimal benchmarking, it is impossible to assess the accuracy of this new method. The benchmarking should be significantly extended and the performance should be compared with the method of Williams et al 2018.

I asked what happens if a neutral model is simulated and I would have hoped for a more quantitative answer than the given one. How often would drivers be detected for various parameter settings when no driver is present? Which strength would be estimated for these false positives? Again a wide range of values should be considered.

I also very much agree with the second reviewer that it is hard to read and interpret this manuscript. While some derivations were moved to the SI, most of the text did not change and hence the clarity did not really improve. The same is actually true for the figures. In five different figures, four different notations are used for panels - why? There is no consistency in label sizes, layouts, style etc which does not help to understand this study.

Line 418-420: The authors claim that determining tumor progression can be useful to understand a tumor's aggressiveness at the time of sequencing. However, doesn't the presented approach determine the growth rate perhaps years before the sequencing? Whether or not the tumor still grows with these dynamics is unclear from the detectable mutations.

Lines 430-435: The authors cited many papers which actually simulated tumors with billions of cells which contradicts that it is infeasible to simulate these tumors. Moreover, using some basic results from population genetics also this model could be simulated much more efficiently and allow larger population sizes.

Lines 441-449: How did the authors infer an effective population size of 1 million? Even early stage tumors have an actual population size of multiple billions of cells. This estimate seems very important for choosing realistic values for the parameter k .

Data access: The authors could simply remove all genomic coordinates and share the data such that their results are reproducible. I think only the gene name, mutation effect, and VAF is needed to reproduce these results.

Reviewer #2 (Remarks to the Author):

The authors have done a good job of responding to the reviewer comments generally and the manuscript has been substantially improved.

The ideas contained in the manuscript represent an important contribution and in my view the manuscript is worthy of publication.

I have a few minor additional comments:

The font sizes in Figure 4a & 4b (and to a lesser extent in some other figures) seem rather small.

I picked up a small number of minor grammatical errors, e.g.

Line 288: Using our model, each mutation i from sample in our database is assigned a potential
289 positive or negative growth value r_i and a driver effect k_i .

Line 489: Let assume a simple 490 population of cancer cells that grows exponentially...

We have now revised our manuscript entitled “Estimating growth patterns and driver effects in tumor evolution from individual samples”. Text in yellow background denotes previous changes for revision #1, while text in grey represents our new changes for 2nd revision. We thank the reviewers for their comments and suggestions.

Reviewer #1 (Remarks to the Author):

R1.1 – Further improving previous analyses’ presentation

Referee Comment	The authors have partially addressed my questions. Nevertheless, the benchmarking should be significantly extended. Figure 2 is a first step but this is far away from a comprehensive evaluation what a reader would expect for a methods paper in Nature Communications. There are no error bars. How many runs were simulated? No statistical tests. Values for all parameters should be provided when essential for understanding a figure.
Author Response	We thank the reviewer for the comments and suggestions. We have now revised our figures, but also included a series of new figures based on our new simulations. We now provide a much more extensive benchmarking based on the reviewer suggestions, including neutral and non-neutral simulations as implemented by the Williams et al 2018 software. Standard deviations and standard error bars are now included in our figures. We apologize for any previous omissions, especially since statistical tests had been performed for all analyses.
Excerpt From Revised Manuscript	- Revised main figure 2i,ii in manuscript

R1.2 – Benchmarking scalar k with Williams simulations

Referee Comment	Because of the rather unusual model setup, there is a lot of uncertainty around the parameter values (in particular k), hence a much large range of values should be explored. With this very minimal benchmarking, it is impossible to assess the accuracy of this new method. The benchmarking should be significantly extended and the performance should be compared with the method of Williams et al 2018.
Author Response	To further benchmark our method, in our revised manuscript we have now included hundreds of new simulations based on Williams et al 2018, including a much wider range of simulated selection coefficient s ($0 < s < 34$), projected coefficients s^* for larger populations based on population genetic models, and drivers with VAFs within different ranges. According to our revised figure 2c our results suggest a high correlation between simulations and predicted values ($r=0.6$). Higher and medium driver VAFs signify better effect prediction, while smaller VAFs further improve driver detectability (absolute median distance from simulated driver) (Figure S11).
Excerpt From Revised Manuscript	- Revised main figure 2iii, iv - Revised supplementary figure S1 - Main 266-299: “Neutral and non-neutral...what we observed.”

	 - Suppl. 388-393: “For our independent set... method’s detectability.” - Suppl. 466-495: “Neutral and non-neutral simulations... ability to detect drivers”
--	---

R1.3 – Simulation of neutral models and false peaks

Referee Comment	I asked what happens if a neutral model is simulated and I would have hoped for a more quantitative answer than the given one. How often would drivers be detected for various parameter settings when no driver is present? Which strength would be estimated for these false positives? Again, a wide range of values should be considered.
Author Response	In our revised manuscript we have now included 140 neutral simulations using the software from Williams et al which were further re-analyzed for a range of population size projections and different hitchhiker window sizes. Overall, peaks that were identified using these neutral simulations show a small overlap with weak drivers (Figure 2c and S1f). Based on our method’s behavior, it is not very meaningful to calculate an expected number of false drivers. Our method cannot distinguish between very weak drivers and neutral tumor progression, but can accurately detect stronger drivers. Instead, we can determine the boundaries of a neutral distribution and define a threshold value for safe prediction. These neutral simulations enable us to calculate an exact threshold for safe predictions as shown in figures 2d and S1g-i. This threshold can vary based on population and hitchhiker window size, thus, we can further use these neutral simulations to tune our model’s parameters as shown in figure S1e (for window and population size). However, even without model tuning, our results do not change qualitatively, as smaller window sizes lead to similar increase in predicted k for neutral or non-neutral simulations, while larger populations (through projection) reduce the standard deviation for the size of neutral peaks, further improving driver detectability. Our results once again suggest that false positives are radically decreased when the predicted effect k or the simulated coefficient s exceed a specific value -- which can be easily determined based on these simulations and for different parameters (figure 2c,d and S1g,h). This value mostly depends on the narrow distribution of neutral effect peaks. We now show how the user can determine this threshold for different populations and various window sizes. In agreement with the reviewer’s suggestion, we can now pinpoint a precise threshold value for safe detection along various model and population parameters. In the figure below, we show how stronger driver predictions decrease the number of false positives above a threshold value.

	Stronger k^* predictions ($>1.17^*$) result in accurate driver detection Abs. median distance (# of ranked muts.) Predicted k effect projected for 1,000,000 cell population size (k^*) More extensive analyses are presented in the manuscript.
Excerpt From Revised Manuscript	 - Revised main figure 2iii, iv - Revised supplementary figure S1 - Main 266-299: “Neutral and non-neutral...what we observed.” - Suppl. 388-393: “For our independent set... method’s detectability.” - Suppl. 466-495: “Neutral and non-neutral simulations... ability to detect drivers”

R1.4 – Manuscript clarity and comprehension

Referee Comment	I also very much agree with the second reviewer that it is hard to read and interpret this manuscript. While some derivations were moved to the SI, most of the text did not change and hence the clarity did not really improve.
Author Response	Before and throughout the reviewing process we have applied excessive editing and changes in our manuscript based on the reviewers’ suggestions. Apart from moving a large section to SI, we have also replaced this section with two sections consisting of a high-level description of our model, which we think is crucial for the comprehension of our method. We have tried to better and further clarify our reasoning behind our methodology and the description of our model.
Excerpt From Revised Manuscript	All changes in manuscript marked with yellow (1 st revision) and grey (2 nd revision)

R1.5 – Consistency between figures

Referee Comment	The same is actually true for the figures. In five different figures, four different notations are used for panels - why? There is no consistency in label sizes, layouts, style etc which does not help to understand this study.
Author Response	In our revised manuscript we have improved our figure style in terms of presentation, consistency (whenever possible) and clarity. Furthermore, we have tried our new analyses to align well with previous notations.
Excerpt From Revised Manuscript	Figure 2i-ii: Added error and $2*\sigma$ bars. Formalized axes names for consistency. Figure 3a,b,c,d: Formalized axes’ names for consistency.

	Figure 4: Formalized axes' names for consistency. Increased font size for clarity Figure 5: Formalized axes' names for consistency in agreement with figure 2 and 3
--	---

R1.6 – Tumor's growth pattern during sequencing

Referee Comment	Line 418-420: The authors claim that determining tumor progression can be useful to understand a tumor's aggressiveness at the time of sequencing. However, doesn't the presented approach determine the growth rate perhaps years before the sequencing? Whether or not the tumor still grows with these dynamics is unclear from the detectable mutations.
Author Response	This is a great correction by the reviewer who is right in pointing out that the VAFs used in these analyses represent the tumor's historic growth pattern and not the exact time of sequencing. For this, a much deeper sequencing would be needed. We have now modified the manuscript accordingly.
Excerpt From Revised Manuscript	Main lines 455-457: "each tumor's historic...in our sample)."

R1.7 – Larger population projections

Referee Comment	Lines 430-435: The authors cited many papers which actually simulated tumors with billions of cells which contradicts that it is infeasible to simulate these tumors. Moreover, using some basic results from population genetics also this model could be simulated much more efficiently and allow larger population sizes. Lines 441-449: How did the authors infer an effective population size of 1 million? Even early stage tumors have an actual population size of multiple billions of cells. This estimate seems very important for choosing realistic values for the parameter k.
Author Response	Although the comment implies that papers we cite directly simulate tumors with at least a billion cells, those we are aware of (and can use the simulated VAF), for instance Williams 2018, simulate a smaller tumor ($\sim 10^4$ cells at sequencing time), and then use population genetics scaling results to predict simulation outcomes on realistic population sizes ($\sim 10^{10}$ cells), which we also did in our previous revision. In our newly revised manuscript, we explored a wider range of larger populations, while we use two approaches to predict the performance of our algorithm on realistic population sizes using simulations. In addition to the population scaling approach, which we adapt from Williams, we also directly substitute a range of realistic population sizes into Eq. 3 and s16 (In Supplement) as the variable N_{tot} to obtain projected k^* values (Figure S1k). As discussed below, both approaches show our algorithm to be robust across a range of population sizes.

	Similar to Williams et al 2018, we use population genetic models to project the selection coefficient s^* for larger population sizes up to 1,000,000,000 cells. In this sense, like previously, we simulated up to 1 billion cells. We agree that the estimate of an effective population size of 1 million may be unrealistic for some tumors (we stated it only as a conservative estimate, since for larger sizes the estimator variance will be reduced). We have thus used a wider range of population sizes when applying the two methods mentioned above in our revised manuscript. As we observe, the use of larger population size estimates for N_{tot} leads to an improvement in the accuracy with which we detect simulated drivers (in terms of the distance from the true driver). In our model, adjusted values k^* can be easily modified in the script code to account for larger populations, simply by providing the population size. In practice, as shown in figure S1j, these projections provide a smaller predicted k^* due to the larger population size, but do not burden driver detectability (which is slightly improved, possibly due to noise reduction and smaller neutral effects). Our scripts provide an easy opportunity for the user to obtain k^* values based on his/her choice of population size.
Excerpt From Revised Manuscript	 - Revised main figure 2iii, iv - Revised supplementary figure S1 - Main 266-299: “Neutral and non-neutral...what we observed.” - Suppl. 47-51: “Projected scalar effect k^*...for larger population sizes.” - Suppl. 388-393: “For our independent set... method’s detectability.” - Suppl. 452-454: “We consider a range... in many tumors).” - Suppl. 459-495: “As an alternative to scaling... ability to detect drivers”

R1.8 – Data access restrictions

Referee Comment	Data access: The authors could simply remove all genomic coordinates and share the data such that their results are reproducible. I think only the gene name, mutation effect, and VAF is needed to reproduce these results.
Author Response	We understand the reviewer’s frustration, In the revised version we have provided gene lists and mutation types as additional material. We note also that PCAWG by its nature is a resource project and that all the data -eg VAFs for mutation coordinates- are available under protected access. The reader can apply for data permission through the regulated process we already mention in the supplement.
Excerpt From Revised Manuscript	 - Suppl. 539-546: “PCAWG state-of-the-art...variance information have been randomly altered”

R2.1 – General assessment

Referee	The authors have done a good job of responding to the reviewer comments
---------	---

Comment	generally and the manuscript has been substantially improved.
Author Response	We thank the reviewer for the positive response and overall feedback throughout the process.
Excerpt From Revised Manuscript	NA

R2.2 –Figure 4 font sizes

Referee Comment	The font sizes in Figure 4a & 4b (and to a lesser extent in some other figures) seem rather small.
Author Response	We have now increased the font size for these figures.
Excerpt From Revised Manuscript	NA

R2.3 –grammatical errors

Referee Comment	I picked up a small number of minor grammatical errors, e.g. Line 288: Using our model, each mutation i from sample in our database is assigned a potential 289 positive or negative growth value r_i and a driver effect κ_i . Line 489: Let assume a simple 490 population of cancer cells that grows exponentially...
Author Response	We have now revised the specific lines
Excerpt From Revised Manuscript	NA

REVIEWERS' COMMENTS:

Reviewer #1 (Remarks to the Author):

The authors have addressed all my comments. I think that the results are now much more clear and the paper overall became better. The pseudo VCF files are very much appreciated. I still think that the presentation could be improved some more but that is minor. In particular, the figures do not compare to other papers in Nature Communications. For example, I don't understand why there need to be so many different font sizes in the same figure. Would be easy to fix.

Reviewer #1 (Remarks to the Author):

R1.1

Referee Comment	The authors have addressed all my comments. I think that the results are now much more clear and the paper overall became better. The pseudo VCF files are very much appreciated.
Author Response	We thank the reviewer for helping us improve the manuscript
Excerpt From Revised Manuscript	N/A

R1.2

Referee Comment	I still think that the presentation could be improved some more but that is minor. In particular, the figures do not compare to other papers in Nature Communications. For example, I don't understand why there need to be so many different font sizes in the same figure. Would be easy to fix.
Author Response	In our revised manuscript we have further improved our figures
Excerpt From Revised Manuscript	We have further modified figures 2,3 and 4 for more consistency and readability, including font sizes and colour enhancement.

Reviewer #2 (Remarks to the Author):

N/A